



# ERA5-Land: A state-of-the-art global reanalysis dataset for land applications

Joaquín Muñoz-Sabater[1], Emanuel Dutra[2,3], Anna Agustí-Panareda[1], Clément Albergel[4,5], Gabriele Arduini[1], Gianpaolo Balsamo[1], Souhail Boussetta[1], Margarita Choulga[1], Shaun Harrigan[1], Hans Hersbach[1], Brecht Martens[6], Diego G. Miralles[6], María Piles[7], Nemesio J. Rodríguez-Fernández[8], Ervin Zsoter[1], Carlo Buontempo[1], and Jean-Noël Thépaut[1]

[1]European Centre for Medium-range Weather Forecasts, Reading, UK
[2]Instituto Português do Mar e da Atmosfera, Lisbon, Portugal
[3]Instituto Dom Luiz, IDL, Faculty of Sciences, University of Lisbon, Lisbon, Portugal
[4]CNRM, Université de Toulouse, Météo-France, CNRS, Toulouse, France
[5]European Space Agency Climate Office, ECSAT, Didcot, UK
[6]Hydro-Climate Extremes Lab (H-CEL), Ghent University, Ghent, Belgium
[7]Image Processing Laboratory (IPL), Universitat de València, València, Spain
[8]Centre d'Etudes Spatiales de la Biosphère (CESBIO), Université Toulouse 3, CNES, CNRS, INRAE, IRDe, Toulouse, France

**Correspondence:** Joaquín Muñoz-Sabater (joaquin.munoz@ecmwf.int)

**Abstract.** Framed within the Copernicus Climate Change Service (C3S) of the European Commission, the European Centre for Medium-Range Weather Forecasts (ECMWF) is producing an enhanced global dataset for the land component of the $5^{th}$ generation of European ReAnalysis (ERA5), hereafter named as ERA5-Land. Once completed, the period covered will span from 1950 to present, with continuous updates to support land monitoring applications. ERA5-Land describes the evolution

of the water and energy cycles over land in a consistent manner over the production period, enabling the characterisation of trends and anomalies. This is achieved through global high resolution numerical integrations of the ECMWF land surface model driven by the downscaled meteorological forcing from the ERA5 climate reanalysis, including an elevation correction for the thermodynamic near-surface state. ERA5-Land shares with ERA5 most of the parametrizations that guarantees the use of the state-of-the-art land surface modeling applied to Numerical Weather Prediction (NWP) models. A main advantage of

ERA5-Land compared to ERA5 and the older ERA-Interim is the horizontal resolution, which is enhanced globally to 9 km compared to 31 km (ERA5) or 80 km (ERA-Interim), whereas the temporal resolution is hourly as in ERA5. Evaluation against independent in situ observations and global model or satellite-based reference datasets shows the added value of ERA5-Land in the description of the hydrological cycle, in particular with enhanced soil moisture and lake description, and an overall better agreement of river discharge estimations with available observations. However, ERA5-Land snow depth fields present a

mixed performance when compared to those of ERA5, depending on geographical location and altitude. The description of the energy cycle shows comparable results with ERA5. Nevertheless, ERA5-Land reduces the global averaged root mean square error of the skin temperature, taking as reference MODIS data, mainly due to the contribution of coastal points where spatial resolution is important. Since January 2020, the ERA5-Land period available extends from January 1981 to near present, with 2 to 3 months delay with respect to real-time. The segment prior to 1981 is in production, aiming to a release of the whole



dataset in summer 2021. The high spatial and temporal resolution of ERA5-Land, its extended period, and the consistency of the fields produced makes it a valuable dataset to support hydrological studies, to initialise NWP and climate models, and to support diverse applications dealing with water resource, land and environmental management.

The full ERA5-Land hourly (Muñoz-Sabater, J., 2019a) and monthly (Muñoz-Sabater, J., 2019b) averaged datasets presented in this paper are available through the C3S Climate Data Store at https://doi.org/10.24381/cds.e2161bac and https://doi.org/10.

24381/cds.68d2bb30, respectively.

## 1  Introduction

The land surface state plays a crucial role in the coupled Earth system, especially on seasonal to inter-seasonal predictability and climate projections (Koster et al., 2004). The development of land surface models has greatly benefited from offline

simulations to isolate the role of different land surface processes and to increase the performance of hydrological and thermo-dynamic variables. Land surface models were initially used in offline mode for model development with early inter-comparison studies driven by either in situ observations (Henderson-Sellers et al., 1995; Etchevers et al., 2004) or global reanalysis datasets (Dirmeyer et al., 1999). More recently, multi-model inter-comparison studies and datatsets focusing on water resources monitoring (Harding et al., 2011; Schellekens et al., 2017), or climate modeling (van den Hurk et al., 2016; Krinner et al., 2018)

have gained significant visibility. Offline simulations remain attractive owing to their computational affordability and the needs that follow from the rapid evolution of land surface models (Pitman, 2003).

A key advantage of using offline land surface estimates is their temporal consistency, unlike in the case of coupled land-atmosphere predictions (e.g. operational weather forecasts) that experience frequent updates. Atmospheric reanalyses also provide such a consistency. However, atmospheric reanalysis can be affected by systematic biases, in particular in precipita-

tion, which has led to the development of bias correction methodologies (Weedon et al., 2011; Reichle et al., 2017). Land Data Assimilation systems (LDAS) also provide an important component of reanalyses, which can mitigate model errors and enhance the representation of the land surface state in regions and periods with available observations (Albergel et al., 2017). However this can also result in temporal and spatial inconsistencies (e.g. due to changing observations availability) as well as limitations in the closure of the surface water budget (Zsoter et al., 2019). Examples of existing global offline datasets

are the Global Offline Land-surface Data-set (GOLD) (Dirmeyer and Tan, 2001), MERRA-Land (Reichle et al., 2011) and ERA-Interim/Land (Balsamo et al., 2015). The latter was motivated by important updates to the European Centre for Medium-Range Weather Forecasts (ECMWF) land surface scheme introduced in the operational forecasting model in 2006, when the production of ERA-Interim started. These changes embedded in ERA-Interim/Land provided seasonal forecasting with more accurate and consistent land initial conditions.





Compared to atmospheric reanalysis, offline land surface reanalyses can be produced faster and at more affordable computational cost. An arguable disadvantage compared to high-resolution Earth Observation data is the lack of small scale heterogeneity found in offline model-only based estimates. However, observational datasets suffer from temporal and spatial gaps, and only a few land variables are directly 'observable', so complex algorithms that blend observations and model output are needed to retrieve a complete estimate of land variables. Furthermore, advances in land surface modeling and the increase of

computational resources make it now feasible to run offline models at finer resolutions than traditionally possible. Therefore, offline global simulations are a valuable way to ensure continuity and completeness of the land surface fields, which are two important aspects to foster research at continental scales in climate studies.

       Although the development of offline model estimates has been motivated mainly by climate and weather research, new user requirements are constantly emerging in society. The effects of climate change are pushing different economic sectors

to implement novel adaptation strategies to adjust to the new reality. For example, crop production is already being affected by increasing temperatures and decreased water availability (Lobell et al., 2011; Wheeler and von Braun, 2013; Zhao et al., 2017). This may induce changes to traditional watering crop strategies, harvesting periods, pest management or even crop culture replacements. Likewise, insurance (and reinsurance) companies need reliable historical data to assess the risk of severe droughts or flooding (Tadesse et al., 2015; Jensen and Barrett, 2016), in particular for smallholder agriculture. For public and

private stakeholders, offline land surface datasets can provide complementary information needed to support decision makers.

       This paper documents the new ERA5-Land dataset (Muñoz-Sabater, J., 2019a). Unlike ERA-Interim/Land that was produced as a one-off single simulation research dataset covering the period 1979-2010, ERA5-Land is now an integral and operational component of the Copernicus Climate Change Service (C3S). This means, among others, that the production is guaranteed with timely updates and synchronized with ERA5 monthly updates. To investigate the added value of ERA5-Land, the latter was

compared to two other operational reanalysis products: ERA5 and ERA-Interim (operational until 31 August 2019). This comparison also makes it possible to study the evolution of ECMWF operational reanalyses. ERA-Interim/Land was not included in the comparison as it is only available until 2010 and it was intended to be a research dataset. As reference for the evaluation, in situ observations from different networks around the globe have been used for comparison to the reanalyses estimates. In addition, complementary global gridded model- or satellite-based datasets have been incorporated into the evaluation exercise.

Improvements in ERA5 compared to ERA-Interim are mainly due to 10 years of additional R&D in the use of satellite data in NWP and atmospheric modeling (Hersbach et al., 2020). Differences between ERA5 and ERA5-Land are not so obvious. They both share quite similar parametrizations of land processes, the main improvement of ERA5-Land is due to the non-linear dynamical downscaling with corrected thermodynamic input.

       Section 2 describes the main steps of the methodology used to produce ERA5-Land, while in section 3 the data used

to investigate the added value of ERA5-Land compared to ERA5 and ERA-Interim (mainly from the years 2000-2018) are described. Section 4 shows the results of the evaluation exercise, whereas a discussion of the results and the conclusions are presented in section 5, followed by perspectives for future updates in section 6. Finally, information on how to access the data is presented in section 7.

Earth System
Science
Data

## 2  Methodology

ERA5-Land produces a total of 53 variables describing the water and energy cycles over land, globally, hourly and at a spatial resolution of 9 km. The production is conducted in three segments or streams. The reason is twofold: 1) it allows the production of parallel streams, therefore accelerating the production and the public availability of the data, 2) the atmospheric forcing necessary to produce ERA5-Land is derived from ERA5, thus the production needs the corresponding segment of ERA5 completed for the same time period. Figure 1 shows the different data streams designed in the production of ERA5-

Land. The production started with data from the year 2001 (stream-1) aiming at making available firstly the most recent data, while the back-extension from 1950 to 1980 (stream-3) is currently under production. Each segment or stream is initialized

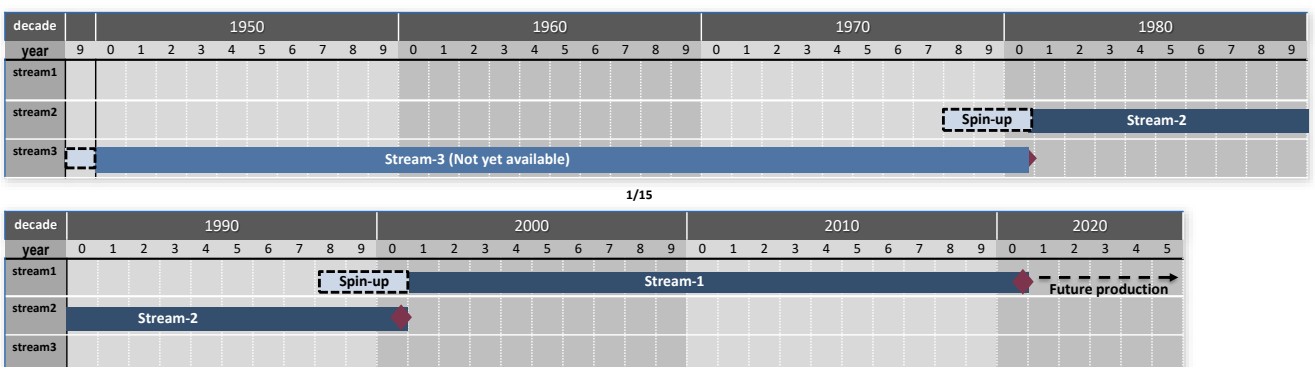

**Figure 1.** Diagram of the production streams of ERA5-Land. The dark blue lines correspond to the data that have already been produced and are available through the C3S Climate Data Store. The light blue line corresponds to the back-extension period and at the time of writing this paper is under production. The three-year spin-up period for the streams 1 and 2 are presented with dashed rectangles. Stream-3 has a one year spin-up (1949). The red diamond presents the end of each stream.

with meteorological fields from ERA5. ERA5-Land does not assimilate observations directly. The observations influence the land surface evolution via the atmospheric forcing. Forcing air temperature, humidity and pressure are corrected using a daily lapse rate derived from ERA5. After that, the land surface model is integrated in 24 h cycles providing the evolution of the

land surface state and associated water and energy fluxes. In addition to the hourly data, monthly means are also computed (Muñoz-Sabater, J., 2019b). Figure 2 shows a diagram of the algorithm used for each 24 h production cycle. The most important components of the production algorithm are presented in the following subsections.

### 2.1  Initialization

ERA5-land is not produced as a single continuous simulation for the entire period. The production is conducted in three

independent streams as shown in Figure 1. To avoid or minimize discontinuities between streams, a careful initialization procedure is needed for each of them. Particular attention must be given to variables carrying long memory. As an example, Figure 3 shows time series of deep soil moisture in a band of latitude between 20° S and 60° S, where the averaged annual



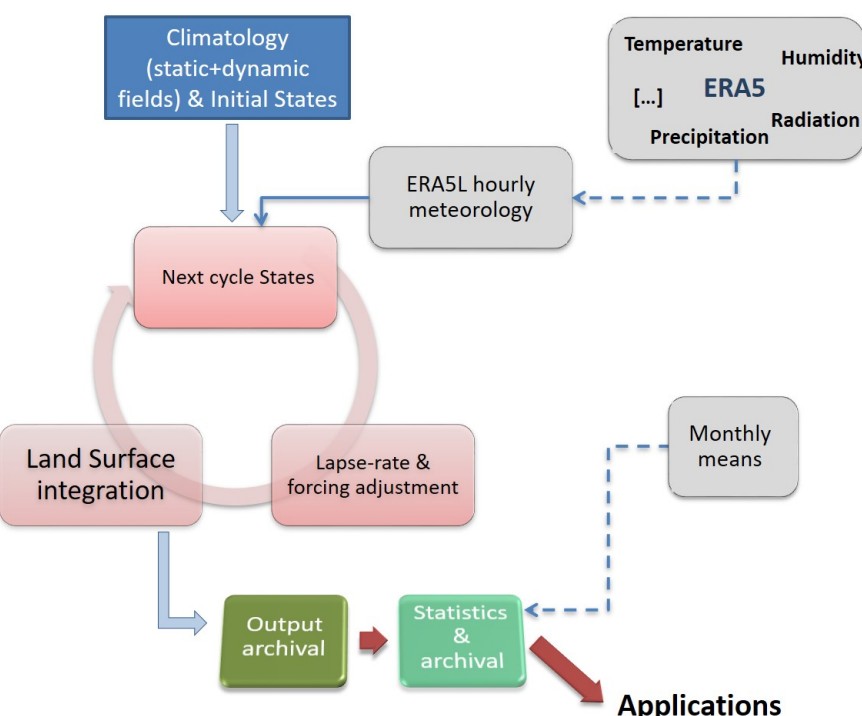

**Figure 2.** Diagram of the algorithm used in the production of ERA5-Land. The land surface model is integrated in 24 h cycles, using short-forecast meteorological forcing fields from ERA5.

soil moisture variability is low. This period includes several production streams of ERA5. ERA5 initialized each stream with ERA-Interim soil moisture initial conditions, which has a different climatology than ERA5. In ERA5 one-year spin-up was

used for each production stream and it is normally long enough for atmospheric variables. However it is not sufficient for deep soil moisture to reach equilibrium, what leads to discontinuities between two production streams, as shown in Fig. 3. The strategy followed to initialize the ERA5-Land production stream starting in 2001 (stream-1) was to use the latest year of a prior long ERA5 stream, and letting three further spin-up years to allow a long spin-up period (see Fig. 1). While this strategy provides satisfactory results for most continental masses around the world, discontinuities are still possible at areas with very

low variability of soil moisture (deserts and polar regions). Particular attention was given to the treatment of permanent snow covered regions. The current model formulation (as in ERA5) does not have an independent treatment of glaciers. Grid-points with glaciers are assigned with a constant snow mass of 10 m. ERA5-Land streams are initialized on the $1^{st}$ of January and a glacier mask is applied to snow mass to guarantee the correct spatial representation of glaciers. A threshold of 50% of a grid box covered by ice is used, below which the snow depth keeps the value computed by the snow scheme of the land model. Values

above the threshold assign a snow water equivalent value of 10 m. This condition is used to avoid grid-points near glaciers with large unrealistic snow depth that result from the interpolation from ERA5 fields to ERA5-Land. For the stream starting in 1981 (stream-2), no prior long ERA5 stream was available. The strategy in this case was to initialize using a ERA5-Land

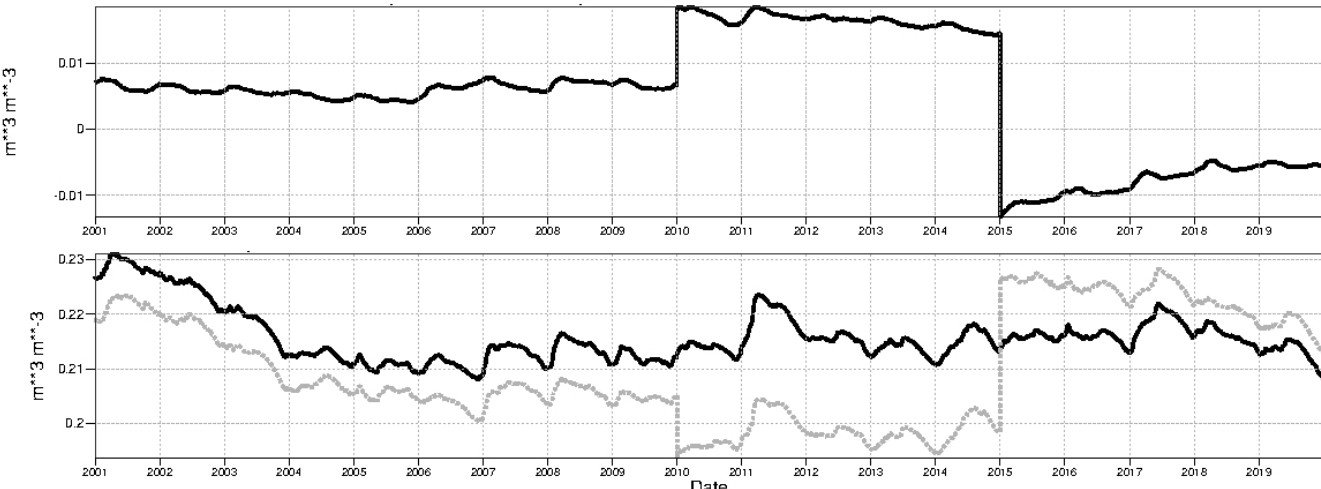

**Figure 3.** Top: Mean differences between ERA5-Land and ERA5 soil moisture time series for the fourth soil layer of the CHTESSEL land surface model (100-289 cm), averaged for the latitudes 60 S - 20 S. The temporal resolution is 12 hours. Bottom: Raw time series of ERA5 (dashed light grey curve) and ERA5-Land (dark curve).

climatology of the $1^{st}$ of January for the period 2001-2018, and then letting three spin-up years. A similar strategy was used to initialize the stream starting in 1950 (stream-3), but in this case a 1981-2010 climatology was used and only one spin-up year
was feasible. The latter was limited by the availability of forcing data.

## 2.2 Static and climatological fields

As in ERA5, the land characteristics are described using several time invariant fields. These consist of the land-sea-mask, the lake cover and depth, the soil type, and vegetation type and vegetation cover. In addition surface albedo and leaf area index are prescribed as monthly climatologies. The complete list of time invariant fields with their information source is provided in
supplementary table S1.

## 2.3 Atmospheric Forcing

ERA5-Land is driven by atmospheric forcing derived from ERA5 near-surface meteorology state and flux fields. The meteorological state fields are obtained from the lowest ERA5 model level (level 137), approximately 10 meters above the surface and include: air temperature, specific humidity, wind speed and surface pressure. The surface fluxes include downward shortwave
and longwave radiation and liquid and solid total precipitation. These fields are interpolated from the ERA5 resolution of about 31 km to ERA5-Land resolution of about 9 km via a linear interpolation method based on a triangular mesh. The atmospheric forcing built for ERA5-Land is hourly and consistent over all the production period, and is the result of the assimilation of a large number of conventional and satellite observations through a 4-Dimensional Variational assimilation System (4D-VAR)





and Simplified Extended Kalman Filter (SEKF) systems as described in Hersbach et al. (2020). Previous land reanalyses have
included corrections to the precipitation forcing to address limitations of the precipitation fields of the atmospheric reanalysis.
This is not the case in ERA5-Land mainly due to the (1) enhanced quality of ERA5 precipitation when compared with previous
atmospheric reanalyses (e.g. Beck et al., 2019; Tarek et al., 2020; Nogueira, 2020) and to (2) reduce dependencies on external
data that would limit the near-real time data availability. However, air temperature, humidity and pressure are corrected for the
altitude differences between ERA5 and ERA5-land grids. This correction involves 4 steps: (i) relative humidity is computed
from interpolated, uncorrected fields; (ii) air temperature is adjusted for the altitude differences using a daily environmental
lapse rate (ELR) field derived from ERA5 lower troposphere temperature vertical profiles (Dutra et al., 2020); (iii) surface
pressure is corrected for the altitude differences and correction of temperature and (iv) specific humidity is computed using the
corrected temperature and pressure assuming that there is no change in relative humidity. Dutra et al. (2020) present a detailed
evaluation of this methodology comparing the use of a constant (time and space) ELR with daily ELR fields derived from
ERA5. This methodology was shown to reduce the mean absolute error (MAE) of daily maximum temperature by 10%, and by
4% for daily minimum temperature with respect to ERA5 when compared with 2941 stations over Western US. The importance
of the ELR correction is shown in the orography map of the Alpine region around Switzerland in Fig 4. ERA5 misses many of
the highest alpine peaks due to the coarser resolution. Taking into account these orographic differences is important for other
land variables such as surface temperature. For instance, the middle and bottom rows show a more realistic spatial pattern of
surface temperature in ERA5-Land, with a clear cold signal over the higher peaks. It is also remarkable to see how ERA5-Land
is able to resolve lakes as the Leman, Neuchatel and Constance. This is especially visible at 06UTC, when the lake surface
temperature is still significantly warmer than the land (Fig 4f).

## 2.4   Land surface model

The core of ERA5-Land is the ECMWF land surface model: the Carbon Hydrology-Tiled ECMWF Scheme for Surface Ex-
changes over Land (CHTESSEL). The main updates with respect to the land component of ERA-Interim are: a) a revised soil
hydrology, introducing an improved formulation of the soil hydrologic conductivity and diffusivity (that now is variable as a
function of soil texture) and surface runoff based on variable infiltration capacity (Balsamo et al., 2010); b) a fully revised
parametrization of the snow scheme, changing the hydrological and radiative properties of the snowpack (Dutra et al., 2010);
c) the introduction of a climatological seasonality of vegetation, in contrast to the fixed vegetation in ERA-Interim (Boussetta
et al., 2013b) d) a new scheme for bare soil evaporation, allowing soil moisture to reach values below the wilting point (Al-
bergel et al., 2012), e) introduction of a lake model to represent the thermodynamics of inland water bodies (Balsamo et al.,
2012) and f) a parametrization that allows the estimation of land carbon fluxes in a modular way with the Jarvis approach that
computes the stomatal conductance without affecting the transpiration components (Boussetta et al., 2013a).

The land surface model version used in ERA5-Land was operational at ECMWF in 2018 with the model cycle Cy45r1.
A detailed description of the model can be found in chapter IV of the Integrated Forecasting System (IFS) documentation
(https://www.ecmwf.int/node/18714, last access December 2020). Compared with the model version of ERA5, the differences
are mostly technical, with the exception of (i) an updated parametrization of the soil thermal conductivity following Peters-

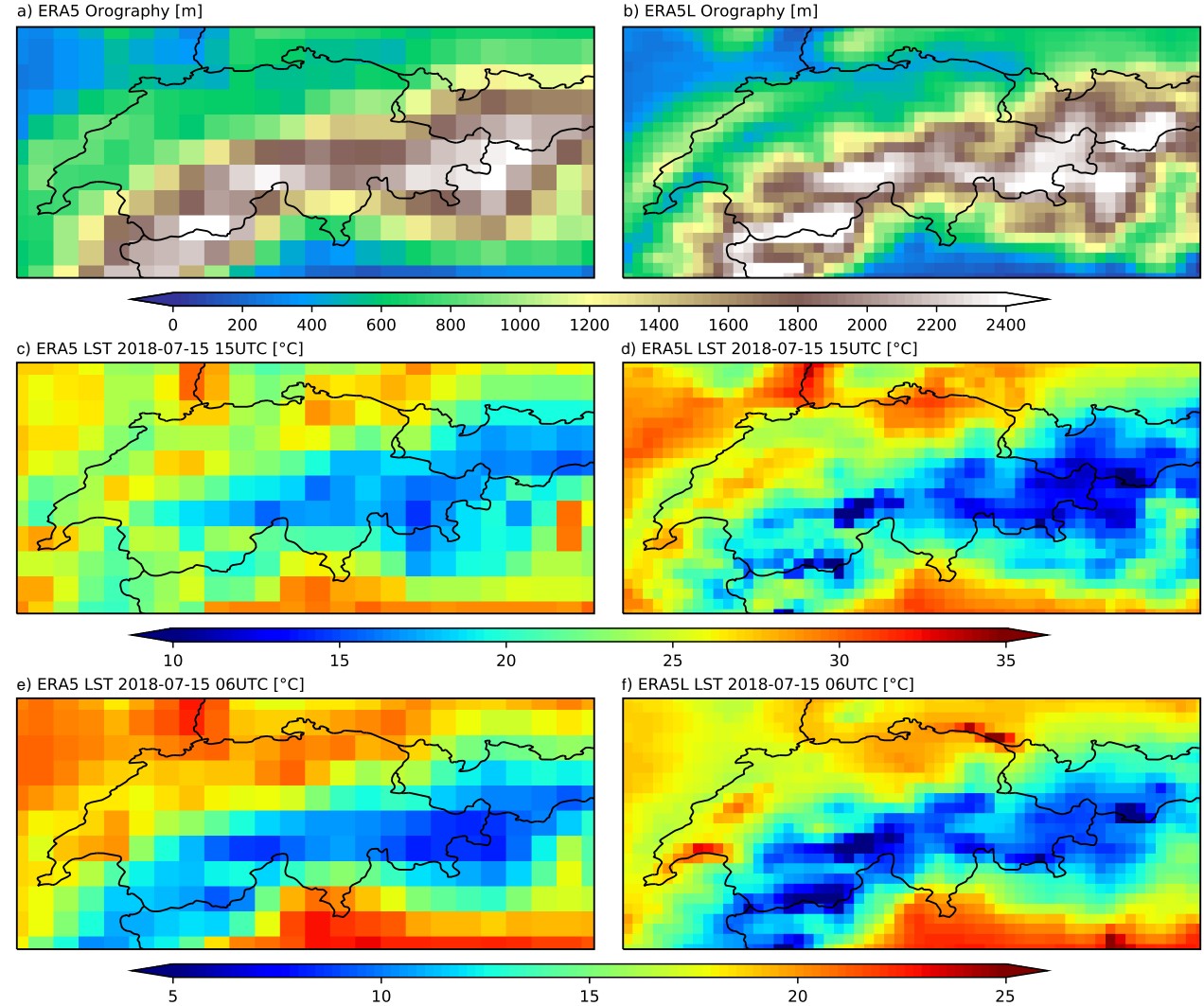

**Figure 4.** ERA5 (left column) and ERA5-Land (right column) orography (top row) of the Alpine region around Switzerland. The middle and bottom rows show the land surface temperature on 15 July 2018 at 15UTC (middle) and 06UTC (bottom), respectively.

Lidard et al. (1998) that takes into account the ice component in case of frozen soil, (ii) a fix to improve conservation for the soil water balance and (iii) rain over snow is accounted for and it does not accumulate in the snowpack. Potential evapotranspiration
(PET) flux in ERA5 suffers from a bug present in IFS cycle 41r2 and that affects PET computation over forests and deserts. This problem has been corrected in ERA5-Land, and unlike in ERA5, ERA5-Land includes PET in the portfolio of products. Given the importance of this variable for some applications, it is worth clarifying that PET is computed by making a second call to the surface energy balance assuming a vegetation type of crops and no soil moisture stress. In other words, evaporation is computed for agricultural land as if it is well watered and assuming that the atmosphere is not affected by this artificial





surface condition. The latter may not always be realistic. Therefore, despite the fact that PET is meant to provide an estimate of irrigation requirements, one has to be cautious especially for arid conditions since the method can give unrealistic results due to too strong evaporation forced by dry air.

## 3  Data and Evaluation strategy

To evaluate the quality of the ERA5-Land fields, several key variables of the water and energy cycles were selected and
compared to available in situ observations and to a series of reference datasets. ERA-Interim and ERA5 reanalyses were included in the comparison, aiming at showcasing the progress of operational reanalyses at ECMWF. The variables evaluated are soil moisture, snow depth, lake surface water temperature and river discharge for the water cycle, the sensible and latent heat fluxes, the Bowen ratio and skin temperature for the energy cycle. This section describes the supporting datasets used in the evaluation exercise and the metrics employed to assess the quality of ERA5-Land fields.

### 3.1  ERA-Interim

ERA-Interim (Dee et al., 2011) is the former ECMWF reanalysis providing estimates for the atmosphere, ocean waves and land surface. It had supported scientific progress during the previous decade and still today is widely used by the scientific community. It includes information on multiple land and atmospheric variables, and it is available from January 1979 to August 2019. It used the IFS version Cy31r2 (more detailed information available at http://www.ecmwf.int/research/ifsdocs/),
corresponding to the IFS 2006 release, with a spatial horizontal resolution of about 80 km and 60 levels in the vertical from the surface to 0.1 hPa. The system includes a 4D-VAR system (Rabier et al., 2000) providing analyses fields at a temporal resolution of 6 hours and 3 hours for short-forecast fields (like for precipitation and fluxes). The main difference between the land component of ERA-Interim and those of both ERA5 and ERA5-Land is that the former is based on TESSEL (van den Hurk et al., 2000), which is considered as the precursor of the current CHTESSEL scheme. In ERA-Interim the soil moisture
and soil temperature analyses are based on a local optimal interpolation scheme (Mahfouf, 1991; Douville et al., 2000) that assimilates SYNOP temperature and relative humidity at screen-level (2 m). The snow depth analysis is independent of the soil wetness and is based on a Cressman analysis that assimilates SYNOP snow reports and snow free satellite observations (Drusch et al., 2004).

### 3.2  ERA5

ERA5 is the latest comprehensive ECMWF reanalysis and has replaced ERA-Interim. It is based on a version of the ECMWF IFS (Cy41r2) that was operational in 2016. ERA5 provides hourly estimates of the global atmosphere, land surface and ocean waves from 1950 and is updated daily with a latency of 5 days. Its state estimates are based on a high-resolution (HRES) component at a horizontal resolution of 31 km and with 137 levels in the vertical spanning from the surface up to 0.01 hPa. Information on uncertainties in these are provided by a 10-member ensemble of data assimilations (EDA) at half the horizontal
resolution. Both the HRES and EDA ERA5 data assimilation use background-error estimates that utilise the output from the



EDA. The land component of ERA5 is, like ERA5-Land, based on the CHTESSEL model, though at a resolution of 31 km rather than 9 km. ERA5 uses new analyses of sea-surface temperature and sea-ice concentration, variations in radiative forcing derived from CMIP-5 specifications, and various new and reprocessed observational data records.

The data assimilation system consists of an incremental 4D-Var component (Courtier et al., 1994) for upper-air and near surface components, an ocean-wave optimal interpolation scheme and a dedicated land data assimilation system (LDAS). The LDAS comprises a two-dimensional optimal interpolation scheme for the analysis of screen-level 2 m temperature and relative humidity, and for snow (depth and density), a point-wise Simplified Extended Kalman Filter (de Rosnay et al., 2013) for three soil moisture layers in the top 1m of soil, and a one-dimensional OI for soil, ice and snow temperature. Details of the ERA5 configuration are described in Hersbach et al. (2020) that also contains a basic evaluation of characteristics and performance

for the segment from 1979 onward. The performance of the component from 1950 to 1978, the back extension which was made available later, is described in Bell et al. (2021) and a detailed analysis for surface temperature and humidity is provided by Simmons et al. (2021). Technical details are also provided in the online documentation (https://confluence.ecmwf.int/display/ CKB/ERA5%3A+data+documentation).

**Table 1.** Overview of the main characteristics of ERA-Interim, ERA5 and ERA5-Land.

|  | ERA-Interim | ERA5 | ERA5-Land |
| --- | --- | --- | --- |
| Period publicly available [(*)] | 1979 - Aug 2019 | 1950 onwards | 1981 onwards<br>(1950-1980, in 2021) |
| Spatial resolution | 79 km / 60 levels | 31 km / 137 levels | 9 km |
| Land Surface Model | IFS (+ TESSEL) | IFS (+ CHTESSEL) | CHTESSEL |
| Model cycle (year) | Cy31r2 (2006) | Cy41r2 (2016) | Cy45r1 (2018) |
| Output frequency | 6-hourly (analyses)<br>3-hourly (forecasts) | hourly | hourly |
| Uncertainty estimate | None | Based on a 10-member 4D-VAR ensemble at 63 km | as for ERA5 |
| Availability behind real time | N/A | 2-3 months (final product)<br>5 days (preliminary product) | 2-3 months (final product)<br>5 days (preliminary product, in 2021) |

[(*)] Availability at the time of submitting this paper

Both ERA5 and ERA5-Land are produced as part of the Copernicus Climate Change Service that ECMWF operates on
behalf of the European Commission and are available from the C3S Climate Data Sore (CDS). ERA5 and ERA5-Land have a large and diverse user base (more than 40,000 users at the end of 2020). Table 1 summarizes the main characteristics of ERA-Interim, ERA5 and ERA5-Land.





### 3.3 Soil moisture

To evaluate the quality of the soil moisture estimates from the various reanalyses, a large number (>800) of in situ sensors in
the period 2010 to 2018, many providing hourly measurements, were used. These sensors belong to the networks that are listed
in Table 2. The networks are located in North America, Europe, Africa and Australia, and all the observations were retrieved
from the International Soil Moisture Network (Dorigo et al., 2011). Three reanalysis soil layers estimates were compared
to measurements by sensors at three different depths. ERA5 and ERA5-Land top layer soil moisture estimates (0-7 cm) were
compared to in situ sensors at 5 cm depth in North America, Africa, Europe and Australia. A more in depth study was performed
in North America, where most of the sensors are located. For this region, surface soil moisture from ERA-Interim was also
evaluated. In addition, ERA5, ERA5-Land and ERA-Interim soil moisture for the second (7-28 cm) and third layer (28-100
cm) were evaluated against in situ measurements at 20 cm and 50 cm depth, respectively. In situ measurements were compared
to the closest grid point of the ERA5, ERA5-Land and ERA-Interim grids if the closest grid point was not farther than the
respective model resolution. In situ time samples were selected in a $\pm 1$ h window with respect to the reanalysis timestamp. If
several observations were retrieved in a single window the averaged was computed. In order to consider an observed time series
suited for the comparison, a minimum of 150 samples was required for the study period. To remove the seasonal cycle, anomaly
time series were also computed. The soil moisture anomaly values at time $t$ ($SM_{AN}(t)$) were computed from the original time
series ($SM(t)$) computing the mean ($\overline{SM}$) and the standard deviation ($\sigma_{SM}$) of soil moisture a +/- 17 days window as follows:

$$SM_{AN}(t) = \frac{SM(t) - \overline{SM}}{\sigma_{SM}} \tag{1}$$

The following metrics were computed between the reanalyses estimates and the in situ time series: the standard deviation
of the difference (STDD), the bias and the Pearson correlation coefficient (R). The latter was also computed for the anomalies
time series ($R_{AN}$). The results were grouped per continent and their distribution presented in box plots. Values are considered
outliers if they are greater than $q_{75} + 1.5 \times (q_{75} - q_{25})$ or less than $q_{25} - 1.5 \times (q_{75} - q_{25})$, with $q_{25}$ and $q_{75}$ the $25^{th}$ and $75^{th}$
percentiles, respectively.

### 3.4 Snow

Snow depth estimates from reanalyses were compared to two sets of observational data. The first one comprises 10 sites
distributed among North America, Europe and Japan. They were selected as reference sites to evaluate cold processes by
models participating in the Earth System Model-Snow Model Intercomparison Project [ESM-SnowMIP] (Krinner et al., 2018;
Ménard et al., 2019). These sites provide benchmarking data for cold processes in maritime, alpine and taiga types of snow
cover and on different types of climates. Table 3 presents these stations. The second dataset is retrieved from the Global
Historical Climatology Network-Daily [GHCN-daily] (Menne et al., 2012b), from 1 July 2010 to 30 June 2018. The version
used is v3.24 (Menne et al., 2012a). This network integrates thousands of land surface stations across the globe. The daily
observed snow depth product was compared to the daily averaged snow depth estimates from ERA-Interim, ERA5 and ERA5-
Land at 00UTC and 12UTC. To compare the reanalysis data with the snow depth observations, the prognostic snow water



**Table 2.** In situ measurement networks used to evaluate soil moisture. The different columns contain the region and the name of the network, the depths of the probes used, and the bibliographic reference to the network.

| Network | Region | Sensor depth used (cm) | Number of sensors used at each depth | Reference |
|---|---|---|---|---|
| SCAN | North America | 5, 20, 50 | 185, 189, 190 | Schaefer et al. (2007) |
| USCRN | North America | 5, 20, 50 | 98, 75, 75 | Bell et al. (2013) |
| SNOTEL | North America | 5, 20, 50 | 290, 300, 297 | Leavesley et al. (2008) |
| SOILSCAPE | North America | 5, 20 | 94, 74 | Moghaddam et al. (2010) |
| TERENO | Europe | 5 | 10 | Zacharias et al. (2011) |
| SMOSMANIA | Europe | 5 | 21 | Calvet et al. (2007) |
| FMI | Europe | 5 | 8 | Ikonen et al. (2016) |
| Remedhus | Europe | 5 | 22 | Martínez-Fernández and Ceballos (2005) |
| Oracle | Europe | 5 | 5 | Tallec et al. (2015) |
| VAS | Europe | 5 | 2 | Lopez-Baeza et al. (2009) |
| HOBE | Europe | 5 | 39 | Bircher et al. (2012) |
| AMMA-Catch | Africa | 5 | 9 | Lafore et al. (2010) |
| DAHRA | Africa | 5 | 1 | Tagesson et al. (2015) |
| Oznet | Australia | 5 | 19 | Young et al. (2008); Smith et al. (2012) |

**Table 3.** List of ESM-SnowMIP sites used for the evaluation of the snow parameters; adapted from Krinner et al. (2018)

| Station | shortname | years | biome | elevation (m) | coordinates |
|---|---|---|---|---|---|
| Col de Porte | cdp | 1994 - 2014 | Alpine | 1325 | 45.30° N 5.77° E |
| Reynolds Mt. East | rme | 1988 – 2008 | Alpine | 2060 | 43.06° N 116.75° W |
| Weissfluhjoch | wfj | 1996 – 2016 | Alpine | 2540 | 46.83° N 9.81° E |
| Swamp Angel | swa | 2005 – 2015 | Alpine | 3371 | 37.91° N 107.71° W |
| Senator Beck | snb | 1995 - 2015 | Alpine | 3714 | 37.91° N 107.73° W |
| Sapporo | sap | 2005 – 2015 | Maritime | 15 | 43.08° N 141.34° E |
| Sodankyla | sod | 2007 – 2014 | Arctic | 179 | 67.37° N 26.63° E |
| Old Aspen | oas | 1997 – 2010 | Boreal forest | 600 | 53.63° N 106.20° W |
| Old Black Spruce | obs | 1997 – 2010 | Boreal forest | 629 | 53.99° N 105.12° W |
| Old Jack Pine | ojp | 1997 – 2010 | Boreal forest | 579 | 53.92° N 104.69° W |

equivalent (SWE, in units of m water equivalent) and snow density ($\rho_{snow}$, kg m$^{-3}$) from the reanalysis were combined to compute the actual snow depth (SD, m):

$$SD = \rho_{water}\, \frac{SWE}{\rho_{snow}}, \tag{2}$$





where $\rho_{water} = 1000\,\mathrm{kg\,m^{-3}}$ is the reference density of the water. The GHCN observations were only considered if snow depth values were positive and missing values were lower than 50% of the recorded time series. Also stations where the snow depth

reported is lower than 1 cm in more than 5% of the total number of days were removed. Finally, stations located on coastal areas with more than 50% of water in the pixel and on permanent snow area (glaciers) were removed. More than 6000 stations passed the quality filters, and their location can be consulted in Fig. 9. The quality of the snow depth from the reanalyses was evaluated at the hemispheric scale by computing the mean bias (here defined as reanalysis estimate minus in situ observation) and Root Mean Square Error (RMSE) for the months between December and June of the 2010-2018 period.

## 3.5 Lakes

In 2015, the CHTESSEL land surface scheme of the operational IFS introduced a lake tile, which represents lakes, reservoirs, rivers and coastal (sub-grid) waters, and is based on the FLake (Fresh-water Lake) model of Mironov et al. (2010). FLake is a one-dimensional model, which uses an assumed shape for the lake temperature profile including the mixed layer (uniform distribution of temperature) and the thermocline (its upper boundary located at the mixed layer bottom, and the lower boundary

at the lake bottom). To run FLake the lake location (or fractional cover), lake depth (most important parameter, preferably bathymetry) and lake initial conditions are required. For the best performance lake depth should be updated with the latest available information to ensure that depths are close to observed values, as overestimated depths can be blamed for cold biases in summer temperatures or lack of ice. State of lakes in FLake is described by seven prognostic variables: mixed-layer temperature, mixed-layer depth, bottom temperature, mean temperature of the water column, shape factor (with respect to the

temperature profile in the thermocline), temperature at the ice upper surface, and ice thickness. In this paper, the lake surface water temperature (LSWT) estimates from the FLake model embedded in ERA5 and ERA5-Land were compared to in situ observations from three different sources of data, during ice-free periods:

- The Alqueva reservoir in Portugal, from the Portuguese University of Evora, providing hourly data from 2017 and 2018. In addition, daily averaged values were also computed,

- Finnish lakes monitored by Finish Environment Institute SYKE, providing daily data (one measurement per day at 8 am local time) from 2000 to 2016. In addition, summer month (June, July and August) average values were calculated,

- Summer months average values from the global inventory "Globally distributed LSWT collected in situ and by satellites; 1985-2009" (https://portal.lternet.edu/nis/mapbrowse?packageid=knb-lter-ntl.10001.3). In total, 348 lakes over the globe with in situ data for 15 years (1995-2009).

Figure 5 provides the location of all the sources of in situ data used in this study. Considering the above observations as the truth, the mean absolute error (MAE) of ERA5 and ERA5-Land LSWT estimates was computed. The significance of these results were tested using the Kruskal–Wallis test by ranks. In addition, the bias distribution was also calculated and presented in 2-D bar graphs. It should be noted that the Alqueva reservoir and the 27 Finnish lakes depths were verified (in situ vs operational values) and are in good agreement; the lake depths provided by the "Globally distributed LSWT collected in situ



and by satellites; 1985-2009" global inventory were only randomly verified (e.g., by comparison with scientific publications) which might add some uncertainty when interpreting the results.

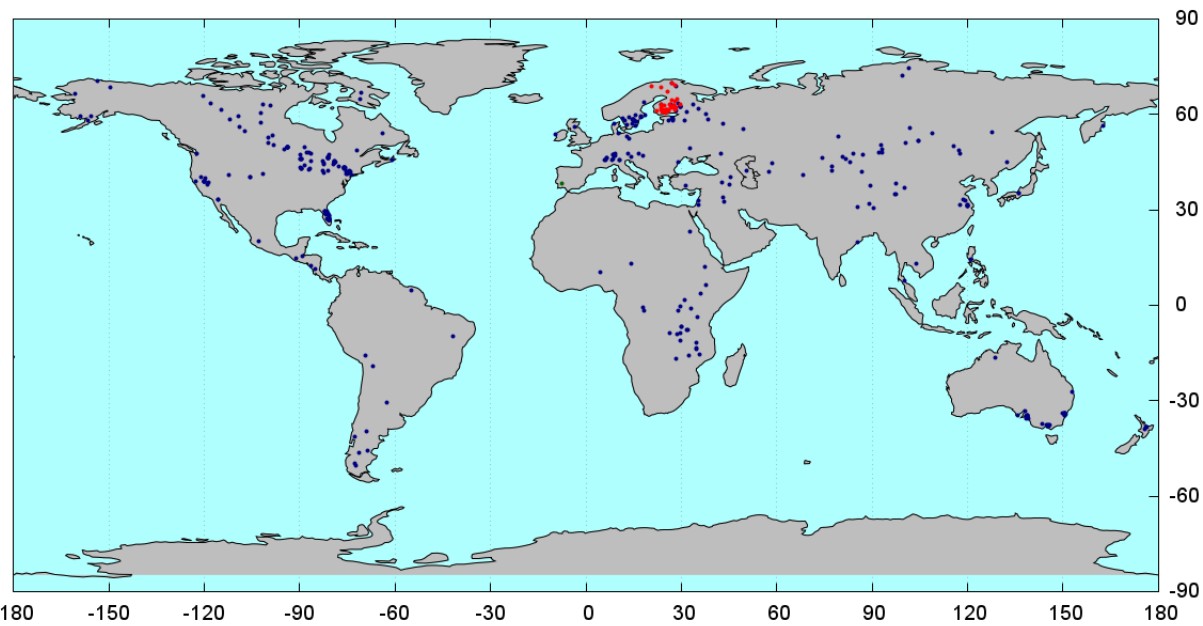

**Figure 5.** Location of lakes with in situ data used in this study; green dots are for hourly and daily data for Alqueva reservoir (Portugal), red for daily and three summer month averaged data (Finnish lakes), blue for three summer month average data for lakes all over the globe.

### 3.6 River discharge

The current version of CHTESSEL does not directly produce river discharge at the river basin scale. Instead, gridded surface and sub-surface runoff from CHTESSEL is coupled to the LISFLOOD hydrological and channel routing model (Van Der Kni-
jff et al., 2010). Coupling ERA5/ERA5-Land runoff with LISFLOOD allows for lateral connectivity of grid cells with runoff routed through the river channel to produce river discharge ($m^3 \cdot s^{-1}$). This is the process used within the Global Flood Aware- ness System (GloFAS; https://www.globalfloods.eu/). More details can be found in Harrigan et al. (2020). River discharge estimates from ERA5 (GloFAS-ERA5) and ERA5-Land (GloFAS-ERA5-Land) were obtained for the period January 2001 to December 2018. This is the common period for which reanalysis data were available at the time of this study. Estimates were
resampled to the GloFAS $0.1°$ gridded river network at a daily time-step. As part of GloFAS a database of global hydrological observations for 2042 stations is held, consisting predominantly (i.e. 75 %) of the Global Runoff Data Centre (GRDC) and supplemented by data collected through collaboration with GloFAS partners worldwide to improve spatial coverage. The loca- tions of the stations have been matched to the corresponding cells on the $0.1°$ GloFAS river network. Following Harrigan et al. (2020) a number of criteria were used to select stations for the evaluation:



– at least 4 years of data available between 2001 and 2018 (not necessarily contiguous),

        – minimum upstream area of 500 km$^2$,

        – difference in catchment area supplied by the data provider and upstream area for the corresponding cell on the GloFAS
          river network must be within 20%,

        – station with the longest record retained when multiple observation stations were matched to the same GloFAS river cell,

In addition to the above conditions, a first order visual quality check on observed river discharge time-series removed stations
     with erroneous data (for example, time series truncated above a threshold, showing several inhomogeneities or series moni-
     toring an artificial canal instead of a river). This filtering procedure resulted in the selection of 1285 stations with drainage
     areas ranging between 575 km$^2$ to $4,664,200$ km$^2$, and a median of $29,963$ km$^2$. Following the methodology of Harrigan
     et al. (2020), hydrological performance was assessed using the modified Kling-Gupta Efficiency ($KGE'$) metric (Gupta et al.,
2009; Kling et al., 2012). The $KGE'$ is an overall summary measure consisting of three components important for assessing
     hydrological dynamics: temporal errors through correlation, bias errors, and variability errors:

$$KGE' = 1 - \sqrt{(R-1)^2 - (\beta-1)^2 - (\gamma-1)^2} \tag{3}$$

$$\beta = \frac{\mu_s}{\mu_o} \qquad\qquad \gamma = \frac{\sigma_s/\mu_s}{\sigma_o/\mu_o} \tag{4}$$

where R is the Pearson correlation coefficient between reanalysis simulations ($s$) and observations ($o$), $\beta$ is the bias ratio, $\gamma$
     is the variability ratio, $\mu$ the mean discharge, and $\sigma$ the discharge standard deviation. The $KGE'$ and its three decomposed
     components (correlation, bias ratio, and variability ratio) are all dimensionless with an optimum value of 1. To evaluate the
     hydrological skill of GloFAS-ERA5-Land, the $KGE'$ can be computed as a skill score, $KGESS$, with GloFAS-ERA5 used
     as the benchmark:

$$KGESS = \frac{KGE'_{GloFAS-ERA5-Land} - KGE'_{GloFAS-ERA5}}{KGE'_{perf} - KGE'_{GloFAS-ERA5}} \tag{5}$$

     where $KGE'_{GloFAS-ERA5-Land}$ is the $KGE'$ value for the GloFAS-ERA5-Land reanalysis against observations, $KGE'_{GloFAS-ERA5}$
     is the $KGE'$ value for the GloFAS-ERA5 benchmark against observations, and $KGE'_{perf}$ is the value of $KGE'$ for a perfect
     simulation which is 1. A $KGESS = 0$ means the GloFAS-ERA5-Land reanalysis is no better than the GloFAS-ERA5 bench-
     mark so has no skill, $KGESS > 0$ when GloFAS-ERA5-Land is considered skilful, and $KGESS < 0$ when the performance
is worse than the GloFAS-ERA5 benchmark.





### 3.7 Energy fluxes

#### 3.7.1 FLUXNET data

The evaluation of the ERA5-Land turbulent fluxes estimates was conducted mostly following the method of Martens et al. (2020). Surface sensible and latent heat fluxes (also denoted in this paper as sshf and slhf, respectively) derived from ERA reanalyses, as well as their ratio (i.e. the Bowen ratio), were compared to measurements from the FLUXNET 2015 synthesis data set (Pastorello et al., 2020). The period under evaluation was based on the availability of reanalysis data at the time of the comparison, and therefore, unlike in Martens et al. (2020), the evaluation period is constrained to 2001-2014. Following Martens et al. (2017), the in situ flux data was subjected to quality control, including (1) the removal of rainy intervals, during which eddy-covariance measurements are typically unreliable, and (2) the removal of gap-filled records. After quality control of in situ stations, only records without gaps during the whole period 2001-2014 were retained, and only sites with a minimum record of five years were retained too. In total, 65 eddy-covariance sites remained after quality control and were used as in situ reference data. These sites are mainly distributed across the continental US, Europe and Australia (see their location in Fig. 17a, b and c, respectively). For each eddy-covariance site, the in situ measurements were aggregated from their native temporal resolution to hourly, 3-hourly and daily intervals. In addition, standarised anomalies were calculated by subtracting for each time interval the climatological expectation (i.e. the average value across the entire record for that interval) and dividing by the standard deviation of that climatology. While the comparison to raw time series may mask the influence of short-term meteorological anomalies on surface energy partitioning (as the temporal variability of turbulent fluxes typically depends strongly on the seasonality of its main drivers), the comparison to anomaly time series reflects the response to short-term meteorological conditions. The Bowen ratio was only calculated at daily temporal resolution for numerical instability reasons. As described in Martens et al. (2020), outliers in the time series of the Bowen ratio, for both the reanalyses and in situ data, were masked using a quantile-based approach.

#### 3.7.2 GLEAM

The Global Land Evaporation Amsterdam Model (GLEAM; Miralles et al. (2011), Martens et al. (2017)) is used in this study with two objectives: a) to compare the GLEAM evaporation estimates directly to the ERA5 and ERA5-Land estimates, which in turn will assess the skill of the underlying land-surface model to simulate turbulent heat fluxes, and b) as an intermediate tool to assess the quality differences of key input meteorological drivers of the turbulent fluxes computed by GLEAM. GLEAM is a process-based, yet semi-empirical, model that computes total evaporation and its separate components over continental masses at global scale. A detailed description of this model can be found in Martens et al. (2017) and Miralles et al. (2010, 2011). In this paper the version 3 (v3) of the GLEAM algorithm is used, and forced with the same database as in the official v3.4a dataset (see www.gleam.eu), including near surface air temperature and surface net radiation from ERA5; this dataset is hereafter referred to as GLEAM+ERA5. Likewise, a version of GLEAM run with ERA5-Land air temperature and surface net radiation is referred to as GLEAM+ERA5-Land. Analogous comparisons between GLEAM+ERA5 and GLEAM+ERA-Interim (using forcing fields from ERA-Interim) can be found in Martens et al. (2020). Surface latent heat flux, surface sensible





heat flux and the Bowen ratio from GLEAM+ERA5 and GLEAM+ERA5-Land were evaluated against the eddy-covariance
data described in section 3.7.1 at a daily time scales.

### 3.8   Skin Temperature

The skin temperature is the theoretical temperature of the Earth's surface that is required to satisfy the surface energy balance.
It represents the temperature of the uppermost surface layer, which has no heat capacity and so can respond instantaneously
to changes in surface fluxes. The top surface layer of the ECMWF's land surface model, CHTESSEL, covers the top 7 cm.
In order to evaluate the skill of ERA5-Land Land Suface Temperature (LST), the NASA's Moderate Resolution Imaging
Spectroradiometer (MODIS) MYD11C3/MOD11C3 version 6 product was used in this study. It provides monthly LST and
emissivity values for Aqua and Terra in a $0.05°$ (5600 meters at the equator) latitude/longitude Climate Modeling Grid (CMG),
for day and night overpasses. Chen et al. (2017) recommended the use of the MODIS LST average ensemble (i.e. from Aqua-
day, Aqua-night, Terra-day, and Terra-night) for climate studies and reported validation with 156 flux tower measurements
(RMSE=2.65, mean bias $< \pm 1$ K). In this study, the MODIS observational LST ensemble was constructed following Chen
et al. (2017) and used as a reference for comparison with ERA-Interim, ERA5 and ERA5-Land monthly averaged LST data
for the period January 2003 to December 2018 (16 years). MODIS data was firstly averaged to monthly time scales and then
upscaled at two spatial resolutions: to $0.1°$ for comparison to ERA5-Land, and to $0.25°$ for comparison to ERA5 and ERA-
Interim data. Bias, Pearson correlation (R), and RMSE were computed on the full time series and correlation was also computed
on the anomalies ($R_{AN}$), calculated as departures from the monthly climatology.

## 4   Evaluation results

### 4.1   Soil moisture

Figure 6 shows the evaluation results for ERA5-Land and ERA5 top soil moisture layer against 5 cm depth measurements
for sites in Europe, Africa and Australia. For sites in Europe, both ERA5-Land and ERA5 show similar STDD and bias
distributions (Figs. 6a, c). The distribution of the R is also similar but the median value obtained for ERA5-Land is slightly
higher than for ERA5 (Fig. 6b). In contrast, ERA5 shows a slightly higher median for the anomaly correlation although
ERA5-Land shows a more compact distribution (Fig. 6d). In Africa only 10 sensors at a depth of 5 cm were available. The
STDD shows a larger distribution for ERA5-Land (Fig. 6e), however the latter shows slightly lower bias and better R (Fig.
6f,g). The anomaly correlation is largely improved (Fig. 6h). Finally, in Australia ERA5-Land boxplots (Figs. 6i-l) clearly
show lower STDD and bias, and higher R than ERA5 (both for the original time series and the anomalies times series). The
evaluation obtained over Europe, Africa and Australia shows an overall slightly better performance of ERA5-Land over ERA5,
in particular the anomaly correlation of the top layer is improved predominantly in the warmest climates. Nonetheless these
results for the top soil layer are not conclusive, partly because of the insufficient number of available stations.



**Figure 6.** Box plots showing the evaluation of ERA5-Land and ERA5 top layer soil moisture against in situ measurements at 5 cm for sites in Europe (top row), Africa (middle row) and Australia (lower row). The first column of panels (left) shows the standard deviation of the difference (STDD), the second column is the Pearson correlation coefficient (R), the third column is the bias and the fourth column is the Pearson correlation of the anomalies time series ($R_{AN}$). On each box, the central mark indicates the median, and the bottom and top edges of the box indicate the 25th ($q_{25}$) and 75th ($q_{75}$) percentiles, respectively. The whiskers extend to the most extreme data points not considered outliers.





To get more insight into the soil moisture evaluation, ERA5-Land and ERA5 were also evaluated over North America, where most of the in situ sensors are available, including several hundreds of sensors at 20 and 50 cm depths. In addition, the same evaluation was performed for ERA-Interim to address the evolution of ERA5 and ERA5-Land with respect to the previous reanalysis generation. Figure 7 shows the evaluation results for ERA5-Land, ERA5 and ERA-Interim soil moisture top three layers against measurements for sites in North America at 5 cm (Figs. 7a-d), 20 cm (Figs. 7e-h) and 50 cm depth (Figs. 7i-l). ERA5 and ERA5-Land surface soil moisture obtains quite similar results for all the metrics, although the median of the correlation and anomaly correlation are slightly better in ERA5-Land. ERA-Interim obtains lower values for these last two metrics, meaning that the land improvements in CHTESSEL compared to the old TESSEL scheme leads to better representation of the soil moisture dynamics. In deeper layers, the performances of ERA5-Land are better than those of ERA5, in particular for the third layer, for which ERA5-Land shows lower STDD (Fig. 7i) and higher correlation (Fig. 7j,l) than ERA5. It is worth mentioning that for the third soil layer of the model, ERA5 performs quite similarly to ERA-Interim, which is likely due to the initialization of the ERA5 streams with ERA-Interim soil moisture conditions.

## 4.2 Snow

Figure 8 shows the mean bias and RMSE of the SWE normalized by the mean value and standard deviation of the observations, respectively, for the 10 sites of Table 3. Each column displays the statistics of the nearest neighbour point, whereas the vertical bars represents the minimum and maximum values of the metrics computed over the four nearest points. Hence, enabling the characterisation of the spatial variability of the errors given that many sites are located in complex terrains or coastal area. Considering the mountain sites, ERA5-Land shows lower RMSE over the sites with moderate altitude, i.e, between 1300 m and 2500 m (cdp, rme, wfj). ERA5-Land also clearly presents the lowest errors in the Arctic site (sod) and in the Boreal forest sites (oas, obs and ojp). Overall, the biases are smaller in ERA5-Land than in ERA5 (and ERA-Interim) at these sites. For the mountain sites, all reanalyses are characterized by a negative bias, which is very likely due to the smoothing of the orography at the resolution of the reanalysis. The higher horizontal resolution of ERA5-Land, compared to ERA5, helps reducing the bias at cdp, rme and wfj, caused by a better orographic representation. However, the agreement with in situ observations at the sites located in very high mountains (snb and swa, located at altitudes greater than 3300 m) is slightly better with ERA5 than with ERA5-Land. It should be noted that the four nearest grid points indicate a much larger spread of errors for ERA5 than for ERA5-Land. Therefore, compensating errors could lead to better performances of ERA5 at the nearest grid point. The maritime site (sap) also shows lower RMSE in ERA5. At this site, the data assimilation can help in adding/removing snow mass for the right reason (snow density is overall well represented at this site, see time series in supplementary Fig. S1, even though the spatial variability is higher in ERA5. On the contrary, at the forest sites (oas, obs, ojp), snow depth assimilation can remove snow mass to compensate for errors in snow density, the latter which are not considered in the assimilation system. Finally, noteworthy are the improvements in the transition between ERA-Interim and ERA5, in particular at Sodankyla (sod), which is caused by improved parametrizations of the snow model introduced between ERA-Interim and ERA5 productions (Dutra et al., 2010).

Figure 9 shows the maps of the RMSE difference between ERA5-Land and ERA5 snow depth estimates when compared





**Figure 7.** Box plots showing the evaluation of ERA5-Land, ERA5 and ERA-Interim against in situ measurements at 5 (top row), 20 (middle row) and 50 cm (lower row) over North America. The first column of panels (left) show the standard deviation of the difference (STDD), the second column in the Pearson correlation, the third column is the bias and the fourth column is the Pearson correlation of the anomalies time series ($R_{AN}$). On each box, the central mark indicates the median, and the bottom and top edges of the box indicate the 25th ($q_{25}$) and 75th ($q_{75}$) percentiles, respectively. The whiskers extend to the most extreme data points not considered outliers.

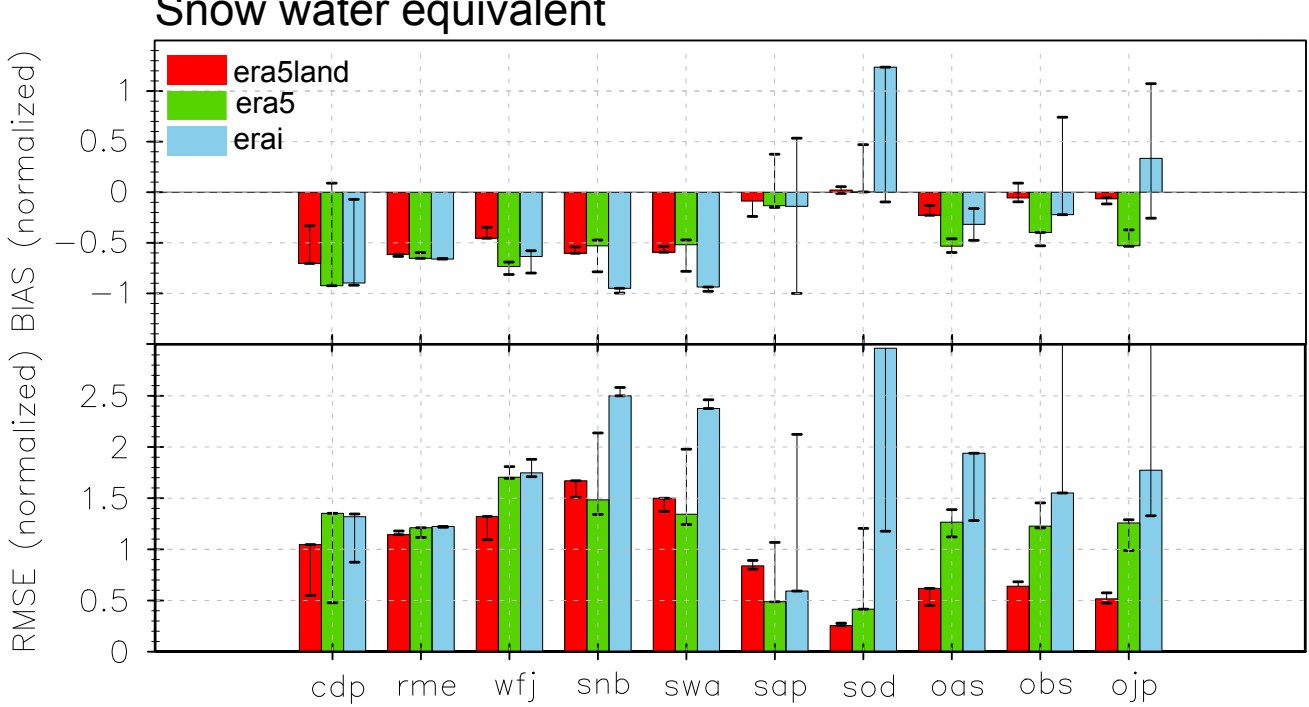

**Figure 8.** Summary statistics of the normalised mean bias (top) and root-mean-square error (RMSE, bottom) of snow water equivalent (SWE) at each ESM-SnowMIP site for ERA5-Land (red), ERA5 (green) and ERA-Interim (cyan). ESM-SnowMIP sites are described in Table 3. Each reanalysis box has a vertical line representing the variability at the 4 nearest grid points to the site location

to in situ observations of the GHCN network, for both North America and Europe. Over the US, and particularly over the Rockies region, ERA5-Land generally outperforms ERA5 in terms of lower RMSE (see Fig. 9a). In these highly complex
terrain regions, the higher horizontal resolution of ERA5-Land adds value by providing more realistic orographic contours. However, over Europe (i.e. mainly Scandinavia, see Fig. 9b) where ERA5 uses a dense SYNOP network of observations in the snow assimilation system, ERA5 performs better than ERA5-Land.

To further quantify the impact of the higher horizontal resolution of ERA5-Land on snow depth simulation, Fig. 10 shows the RMSE as a function of the height of each station. The stations were binned in height ranges of 250 m (0-250, 250-500,
435 etc.) and the distribution of the RMSE is displayed for each height range using boxplots. For heights below ≈ 1500 m, ERA5 performs slightly better than ERA5-Land, while for heights between ≈1500 m and ≈3000 m ERA5-Land outperforms ERA5, as a result of the better resolution. For stations above 3300 m ERA5 performs better, which can be due to compensating errors, similarly to the ESM-SnowMIP sites. In addition, it should be noted that the number of sites at these very high altitudes is small, and therefore the statistical results should be interpreted with caution.



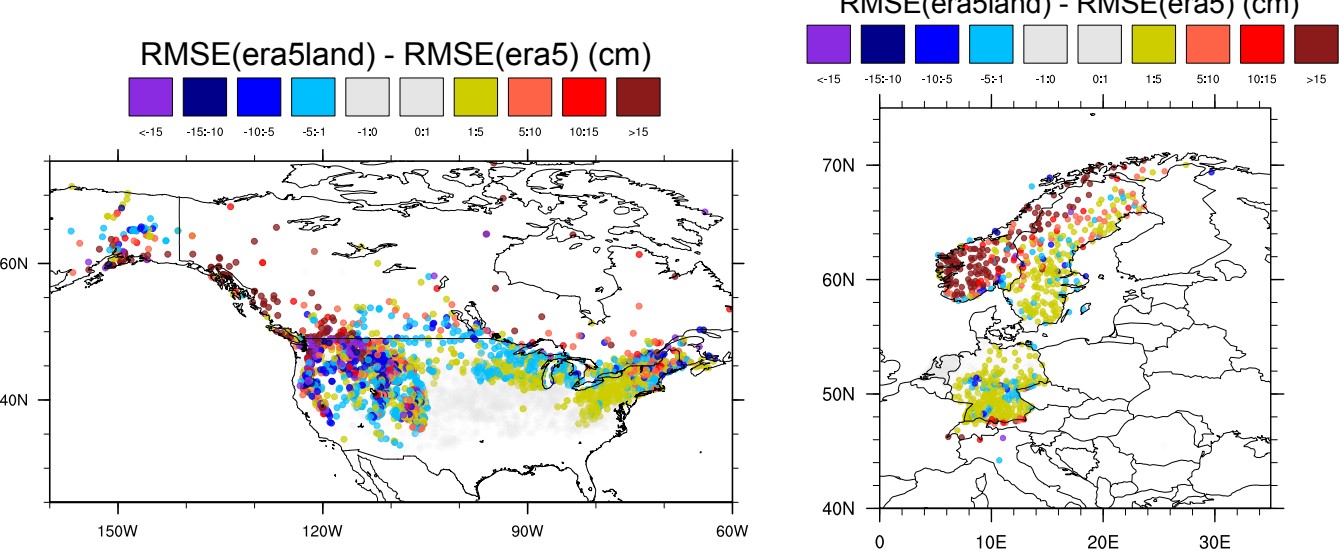

**Figure 9.** Snow depth Root Mean Square Error (RMSE) difference between ERA5-Land and ERA5 (with respect to GHCN observations) for December to June months between 2010 and 2018. Negative values indicate better matching of ERA5-Land averaged snow depth with in situ measurements of the GHCN network, positive values indicate ERA5 averaged snow depth estimates matching better in situ measurements.

## 4.3 Lakes

The ERA5 and ERA5-Land hourly estimates of LSWT compared to the hourly data of the Alqueva reservoir showed that ERA5-Land obtained statistically significant MAE reduction by 2.2 % (see Table 4). While this is a positive result, ERA5-Land data showed some rapid changes in the lake mixed layer depth (up to 5 m) which led to a quick rise of the lake surface water temperature (up to 20°C). Left plot of Fig. 11 shows the bias distribution for hourly data; even though the frequency of bias around zero is larger for ERA5-Land (indicating more accurate estimates of LSWT), errors due to unrealistic temperature rise are also visible at higher frequency between 3-4°C errors.

The comparison of the ERA5 and ERA5-Land daily averaged LSWT with Alqueva reservoir and the 27 Finnish lakes data also showed statistically significant reduction of ERA5-Land MAE (among 28 lakes) by 1.2 % (see Table 4). It should be noted that the lake parametrization scheme relies heavily on the lake depth input data. With the increase of resolution in ERA5-Land, lake depth values change as well. For only those lakes which depth became more realistic in ERA5-Land as result of the higher resolution, the MAE was statistically significantly reduced by 23.8 %; however for only those lakes which depth remained unchanged despite the change of resolution, the MAE was still statistically significantly reduced by 9.6 %, showing the positive influence of higher resolution atmospheric input. The right plot of Fig. 11 shows, for daily data, a reduced amount of large positive errors (cold bias) in ERA5-Land and a larger frequency of errors around zero bias. Finally, only 272 lakes with averaged in situ measurements for the summer months, with 10-15 years-long data records, could be used for comparison with ERA5 and ERA5-Land LSWT estimates. The comparison showed modest statistically significant MAE increase by 1.0 %




**Figure 10.** Boxplot of the snow depth RMSE distribution as function of the site altitude, for North America (30 N - 80 N; 60 W - 160 W, upper panel) and Europe (25 N - 80 N; 10 W - 40 E, lower panel), for ERA5-Land (red), ERA5 (green) and ERA-Interim (cyan). Boxes extend between the lower (25%) and upper quartile (75%) and the horizontal lines within each box represent the median value of the distribution. The black line represents the number of stations grouped at each bin.




**Table 4.** Mean Absolute Errors (MAE) of the ERA5 and ERA5-Land Lake Surface Water Temperature (LSWT) estimates compared to in situ hourly, daily and three summer month average LSWT data. Hourly data correspond to data from the Alqueva reservoir in Portugal (2017 and 2018); daily data are from the Finnish lakes (2000-2016) and averaged hourly data from the Alqueva reservoir; summer month average data are from a global inventory (1985-2009) and the Finnish lakes. First column is for the data type (hourly/daily/summer month average), second colum provides the number of lakes for each category, third colums shows the MAE of ERA5 LSWT estimates compared to in situ measurements, and fourth colum is as third column but for ERA5-Land.

| Data Type | Data amount (number of lakes) | MAE (ERA5) | MAE (ERA5-Land) |
|---|---|---|---|
| Hourly | All (1) | 3.29 | 3.22 |
| Daily | All (28) | 2.71 | 2.68 |
| | Only with unchanged depths; differences due to increased atmospheric horizontal resolution (10) | 2.59 | 2.34 |
| | Only with more realistic depths in ERA5; differences due to decreased surface horizontal resolution (12) | 2.38 | 2.93 |
| | Only with more realistic depths in ERA5-Land; differences due to increased surface horizontal resolution (6) | 3.56 | 2.71 |
| Summer months | All (272) | 3.04 | 3.07 |
| | All, except exceptional lakes (glacier fed, saline and warm lakes) (246) | 2.32 | 2.32 |
| | Non-exceptional lakes with unchanged depths; differences due to increased atmospheric horizontal resolution (107) | 2.22 | 2.25 |
| | Non-exceptional lakes with more realistic depths in ERA5; differences due to decreased surface horizontal resolution (91) | 2.45 | 2.41 |
| | Non-exceptional lakes with more realistic depths in ERA5-Land; differences due to increased surface horizontal resolution (41) | 2.56 | 2.52 |
| | Non-exceptional lakes with unchanged depths that are <50 m deep; differences due to increased atmospheric horizontal resolution (92) | 2.32 | 2.39 |
| | Non-exceptional lakes with more realistic depths in ERA5 that are <50 m deep; differences due to decreased surface horizontal resolution (70) | 2.30 | 2.45 |
| | Non-exceptional lakes with more realistic depths in ERA5-Land that are <50 m deep; differences due to increased surface horizontal resolution (34) | 2.52 | 2.49 |

for ERA5-Land compared to ERA5 (see Table 4). If one bears in mind that FLake was not designed for saline, fed by glaciers or mountains and warm lakes (that excludes 26 exceptional lakes) the MAE of the remaining 246 lakes is similar in ERA5 and ERA5-Land. The right plot of Fig. 12 shows the geographical distribution of the 26 exceptional lakes and their averaged ERA5-Land MAE, which shows large LSWT errors, even beyond 10°C. Note that errors for these lakes are very similar for ERA5. If, as a result of the higher resolution, only lakes which depth became more realistic in ERA5-Land compared to ERA5

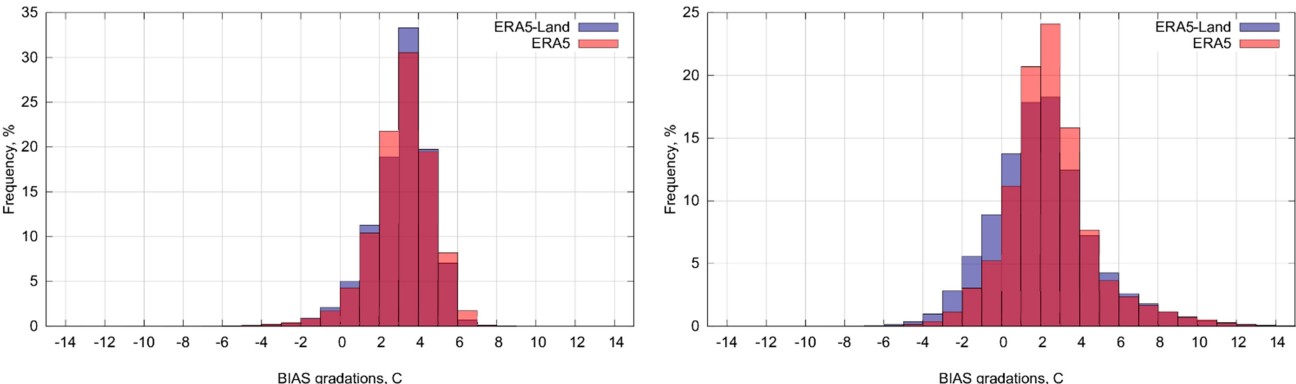

**Figure 11.** Lake Surface Water Temperature bias (observations minus ERA5 or ERA5-Land estimates) distribution (°C) for hourly data from the Alqueva reservoir (left) and daily data from the Alqueva and the Finnish lakes (right). The dark red colour represents the area where both ERA5 (pink colour) and ERA5-Land (violet colour) bias distributions overlap.

are taken into account, the MAE is statistically significantly reduced by 1.4 % in ERA5-Land. FLake was specially designed for medium depth (under 50 m) lakes. Several lakes from this database are however very deep. Nevertheless, computing the statistics for only medium depth lakes did not show any improvement in ERA5-Land data. The left plot of Fig. 12 presents the

bias distribution of non-exceptional lakes (246) that shows wider errors distribution for both ERA5 and ERA5-Land compared to hourly and daily distributions.

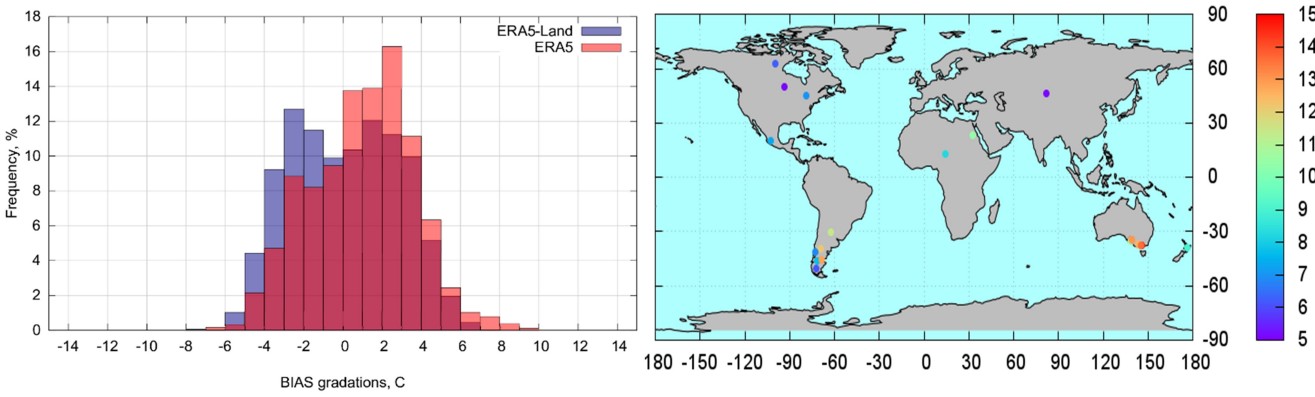

**Figure 12.** Lake Surface Water Temperature bias (observations minus ERA5 or ERA5-Land estimates) distribution (°C) for three summer month average data based on non-exception lakes (left); and geographical distribution of the 26 exceptional lakes (glacier fed, saline and warm lakes) ERA5-Land Mean Absolute Error (right) with colour bar in °C.





## 4.4 River discharge

Results of river discharge performance for GloFAS forced with ERA5 and ERA5-Land are shown in Fig. 13. The overall global median $KGE'$ across 1285 observation stations improves from 0.26 (with an interquartile range of -0.04 to 0.49) for GloFAS-ERA5 to 0.37 (0.08, 0.57) for GloFAS-ERA5-Land (Fig. 13a). In the decomposition of the $KGE'$, the global median Pearson correlation also improves from 0.60 (0.43, 0.75) to 0.64 (0.49, 0.77) (Fig. 13b), and global median bias ratio improves from 0.73 (0.50, 1.13) to 0.89 (0.66, 1.15) (Fig. 13c); equivalent to a 16 % reduction in overall bias. There is very little difference in variability errors between GloFAS-ERA5 and GloFAS-ERA5-Land (Fig. 13d).

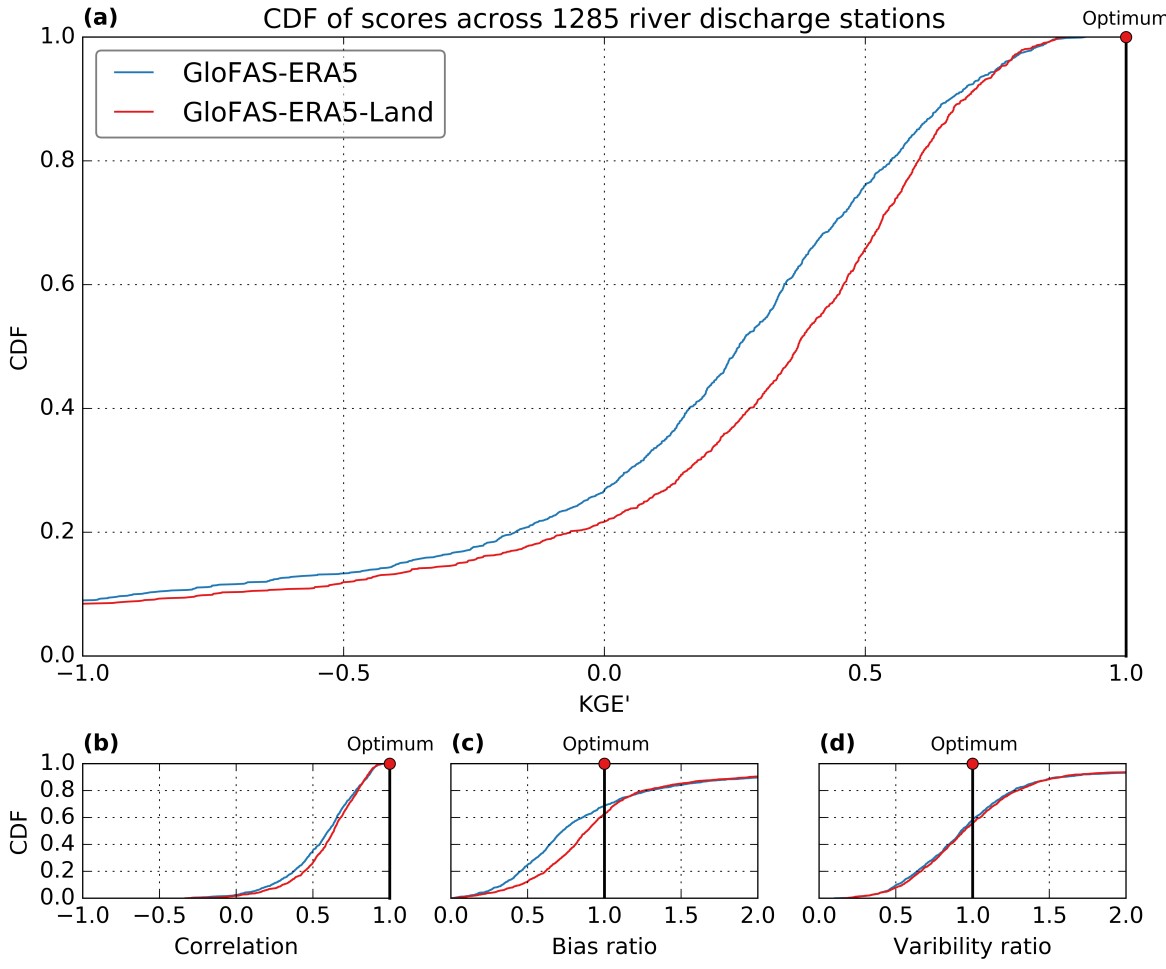

**Figure 13.** Cumulative distribution functions (CDF) of hydrological performance metrics across 1285 observation stations. Modified Kling–Gupta efficiency ($KGE'$) (a) with decomposition of $KGE'$ into Pearson correlation (b), bias ratio (c) and variability ratio (d) for GloFAS-ERA5-Land (red line) and GloFAS-ERA5 (blue line). The red dot marks the optimum value for each metric.

Figure 14 shows the spatial distribution of river discharge skill, measured by the $KGESS$. GloFAS-ERA5-Land shows pos-
itive skill compared to the GloFAS-ERA5 benchmark in 65 % of stations with a global median $KGESS$ of 0.08 (-0.06, 0.25).
Largest improvements in skill are found in North-America, Europe, Northern Russia, Southern Africa and Australian catch-
ments. There is a substantial decrease in skill (i.e. $KGESS$ < -0.2) when forcing GloFAS with ERA5-Land runoff instead of
ERA5 in 11 % of stations, mainly located in Western US and South-America. Care must be taken in spatial representativeness
of these results as the observation network is sparse in some regions of the world, particularly in large parts of Africa and Asia.

**Figure 14.** Modified Kling-Gupta Efficiency Skill Score (KGESS) for GloFAS-ERA5-Land river discharge reanalysis against the GloFAS-
ERA5 benchmark across 1285 observation stations. Optimum value of KGESS is 1. Blue (red) dots show catchments with positive (negative)
skill.

## 4.5 Energy fluxes

### 4.5.1 Evaluation against eddy-covariance sites

The left panel of Fig 15 shows the violin plots of the turbulent fluxes against in situ eddy-covariance measurements for ERA-
Interim and ERA5-Land. The violin plots display three metrics: Bias, i.e., the difference between raw in situ time series and
reanalysis estimates, standardised MAE and anomaly Pearson correlation coefficient ($R_{AN}$).

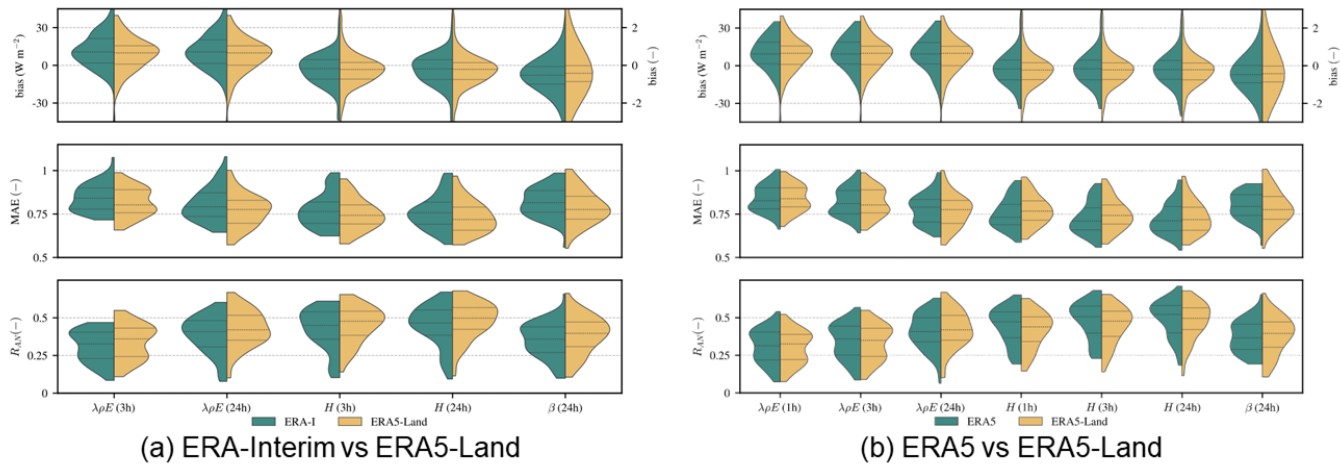

**Figure 15.** Violin plots showing the temporally and spatially averaged statistics of the surface latent heat flux ($\lambda\rho E$), surface sensible heat flux ($H$), and Bowen ratio ($\beta$) from (a) ERA-Interim (green) and ERA5-Land (yellow) and (b) ERA5 (green) and ERA5-Land (yellow). Statistics are calculated with respect to in situ eddy-covariance measurements at both 3-hourly and daily (24h) temporal resolutions. Violin plots represent the distribution of the individual validation statistics with indication of the median and inter-quartile range, and are calculated using a kernel density estimation approach. Statistics include the Bias, Mean Absolute Error (MAE), and the anomaly correlation coefficient ($R_{AN}$). Unlikely most metrics and variables, the scale used to show the bias distribution of $\beta$ is that of the right y-axis on the top row of each panel.

ERA5-Land compares systematically better to in situ measurements than ERA-Interim, with higher $R_{AN}$ values for the heat fluxes and the Bowen ratio and for both sub-daily and daily temporal resolutions, as well as presenting lower bias and MAE. This result is expected, as the land surface model used in ERA5-Land benefits from many improvements compared to that of ERA-Interim (see section 2.4). The right panel of Fig. 15 shows the same plot but comparing ERA5 and ERA5-Land.
The violin plots are much more similar, biases for both fluxes are only marginally better in ERA5-Land (the median of the distribution is nearly the same, but the 75% percentile is always lower), MAE is typically lower for slhf in ERA5-Land (except at 1-hourly resolution), but higher for sshf. On average, correlations for ERA5-Land are only better for slhf at daily resolution and for the Bowen ratio. It also should be noted that all statistics are on average better for sshf than for slhf, both in ERA5 and in ERA5-Land (with lower bias, lower MAE, and higher $R_{AN}$); this matches the results reported by Balsamo et al. (2015)
and Martens et al. (2020). To investigate potential areas of significant improvement/degradation, the left panel of Fig 16 shows the maps of $R_{AN}$ differences of slhf between ERA5-Land and ERA5, over the Continental United States (CONUS). $R_{AN}$ is typically better for ERA5-Land across most US stations, especially near the coasts and around the lakes, where high resolution is more important. In Europe (Fig. 16, middle panel) results are very mixed, but for most of the stations in the Alps ERA5-Land performs worse than ERA5. Although for Australia ERA5-Land is slightly better than ERA5 (see right panel of Fig. 16), it is

not the case for all stations. The results for the Bowen ration are aligned with those of slhf (see supplementary Fig. S3) whereas those of sshf are more favourable for ERA5 (see supplementary Fig. S2).

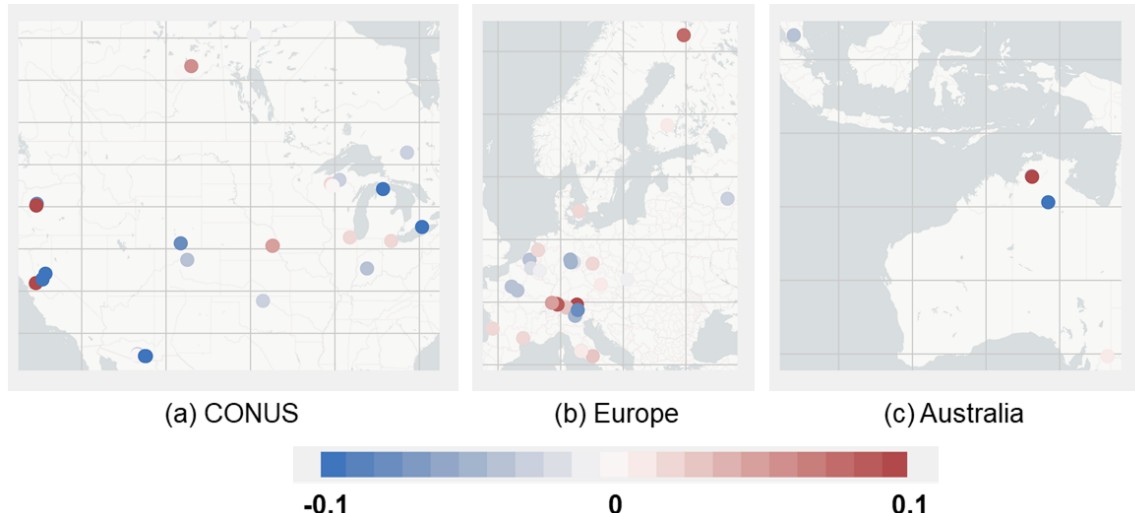

**Figure 16.** Standarised anomaly correlation ($R_{AN}$) difference of the surface latent heat flux between ERA5-Land and ERA5 with respect to eddy-covariance data. Blue colours indicate that the anomaly correlation of ERA5-Land with respect to eddy-covariance measurements is higher than for ERA5, whereas red colours indicate the opposite.

To study the influence of the eddy-covariance sites altitude and air temperature in the heat fluxes, Fig 17 shows the standardized mean $R_{AN}$ (circles) and MAE (squares) of sshf and slhf as a function of the station altitudes and air temperature. The results are in line with those presented in the violin plots. ERA5-Land clearly performs better for slhf and the Bowen ratio,

and the largest differences between ERA5 and ERA5-Land are obtained for the sites located at high altitudes (where more extreme values are obtained and forcing errors are increased), and for the stations where the recorded air temperature is milder. Although for sshf the differences are smaller, ERA5 performs overall better than ERA5-Land and the strongest differences are also seen at high altitude sites.

### 4.5.2   Evaluation using GLEAM

The turbulent fluxes estimated from GLEAM forced with temperature and surface net radiation from ERA5 and ERA5-Land were also computed. ERA5-Land surface fluxes perform better than GLEAM+ERA5-Land when confronted to in situ measurements (see the violin plots in supplementary material Fig. S4), except in term of bias, in agreement with (Martens et al., 2020). This result reflects the quality of the CHTESSEL land surface model, which has greatly improved the parametrization of turbulent energy fluxes and evidences the added value compared to a simpler model as in GLEAM, designed to consider

only input variables that can be observed from satellite. Similar results were obtained when comparing the energy fluxes provided directly by GLEAM and by ERA5-Land (not shown). Fig 18 is similar to Fig 17 but comparing ERA5-Land and



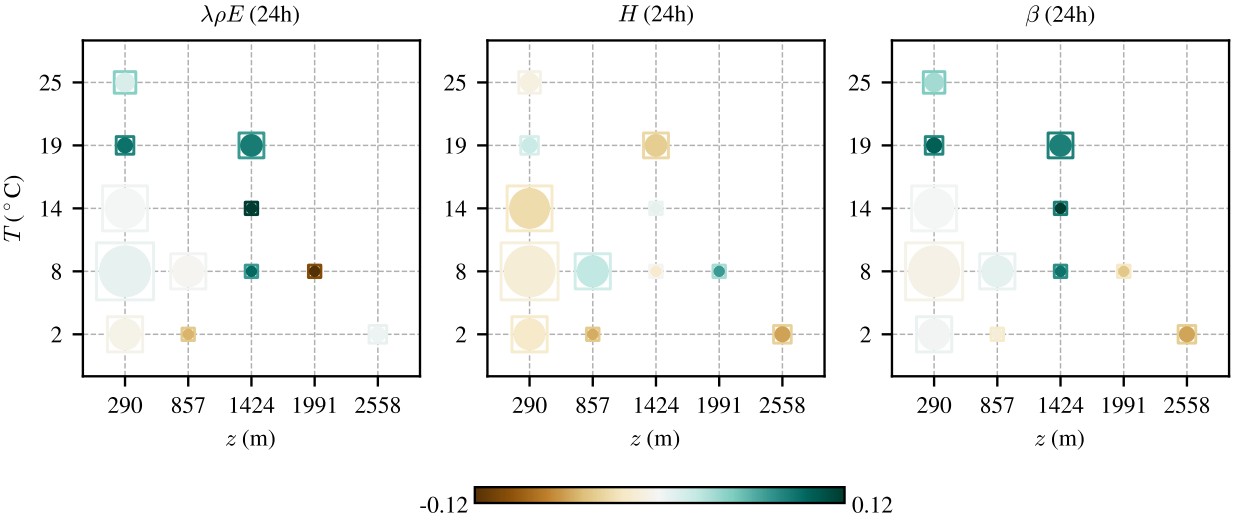

**Figure 17.** Standardised anomaly correlation difference (circles) and standarized mean absolute differences (squares) between ERA5-Land and ERA5 latent heat flux (left plot), sensible heat flux (middle plot) and Bowen ratio (right plot), grouped as a function of stations temperature and altitude. The size of circles and squares is proportional to the number of eddy-covariance towers. Green values denote better matching of ERA5-Land with in situ.

GLEAM+ERA5-Land, as a function of elevation and air temperature, at daily resolution. As expected, almost for all stations temperature and elevation, ERA5-Land heat fluxes match better the in situ reference measurements. The exception are towers at the highest altitudes where forcing errors are generally larger. Comparing GLEAM+ERA5 and GLEAM+ERA5-Land the differences are very small, suggesting that the near-surface air temperature and net radiation are quite similar in both reanalyses for the locations of the eddy-covariance sites.

### 4.6   Skin Temperature

Figure 19 shows global maps of ERA5-Land and ERA-Interim mean LST for the time period 2003-2018, as well as global maps of their correlation and RMSE with respect to the MODIS LST average ensemble, constructed as indicated in section 3.8. While differences in mean LST are apparently minimal among the two simulated products, there is a better correlation of ERA5-Land with MODIS in the tropical band. The low correlation obtained between the simulated and the remotely sensed LST over tropical forests should be taken with caution since persistent cloud cover as well as cloud contamination effects are known to have an impact on remotely sensed observations, leading to important differences in their absolute values (Jiménez-Muñoz et al., 2016; Gomis-Cebolla et al., 2018). Notably, ERA5-Land systematically compares better than ERA-Interim to MODIS LST in terms of RMSE, particularly at high latitudes. As shown in the averaged statistical scores on Table 5, there is an

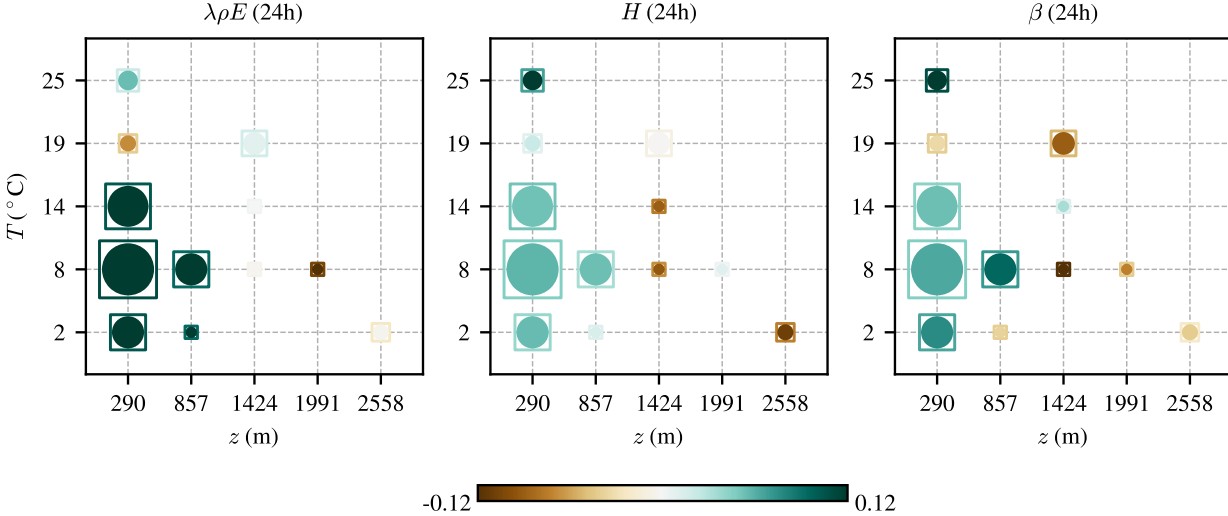

**Figure 18.** Standardised anomaly correlation difference (circles) and standarized mean absolute differences (squares) between ERA5-Land and GLEAM+ERA5-Land latent (left plot), sensible heat flux (middle plot) and Bowen ratio (right plot), grouped as a function of stations temperature and altitude. The size of circles and squares is proportional to the number of eddy-covariance towers. Green values denote better matching of ERA5-Land with in situ.

overall improvement of ERA5 and ERA5-Land LST simulations with respect to ERA-Interim, yet only modest improvements are achieved in ERA5-Land compared to ERA5 LST, mainly on bias and RMSE. The better correlation of MODIS with ERA5 and ERA5-Land than with ERA-Interim is more evident in $R_{AN}$ maps (see Table 5 and Fig.S5 in the supplementary material).

To further scrutinize the impact of the ERA5-Land higher horizontal resolution on LST simulation, ERA5 LST was resampled to $0.1°$ and then global maps of the RMSE difference ($\Delta$RMSE) between ERA5 and ERA5-Land with respect to the MODIS LST average ensemble were calculated. The improvement of ERA5-Land on LST in coastal areas is clearly visible in the $\Delta$RMSE maps, as shown in Fig. 20 for the European domain. The time series at the coastal pixel in Norway illustrates how ERA5-Land LST simulations compare better to the observational record by capturing more closely the annual cycle in its whole range of variability, and therefore decreasing bias and RMSE, while the correlation with the MODIS product is very good and almost similar for both reanalyses. The time series at the other two specific locations illustrate regions of complex topography exhibiting slightly lower RMSE of ERA5-Land (Iceland), and of ERA5 (Alps), with respect to the MODIS LST average ensemble. These results highlight improved LST simulations of ERA5-Land on coastal regions.





**Figure 19.** Mean Land Surface Temperature (LST, in K) in the 2003-2018 period for (a) ERA5, (b) ERA-Interim. Global correlation coefficient (R) maps of (c) ERA5 and (d) ERA-Interim with respect to MODIS LST average ensemble. Global RMSE maps (K) of (e) ERA5 and (f) ERA-Interim with respect to MODIS LST average ensemble. Note that the LST here refers to the skin temperature.

**Table 5.** Spatially and temporally averaged Bias, RMSE, R and Anomaly Correlation Coefficient ($R_{AN}$) between ERA-Interim, ERA5 and ERA5-Land estimates and the MODIS land surface temperature average ensemble for the time period 2003-2018.

|                    | Bias (K) | RMSE (K) | R    | $R_{AN}$ |
|--------------------|----------|----------|------|----------|
| ERA-Interim - MODIS | 3.65     | 5.87     | 0.91 | 0.49     |
| ERA5 - MODIS       | 1.64     | 3.96     | 0.94 | 0.75     |
| ERA5-Land - MODIS  | 1.36     | 3.78     | 0.94 | 0.75     |



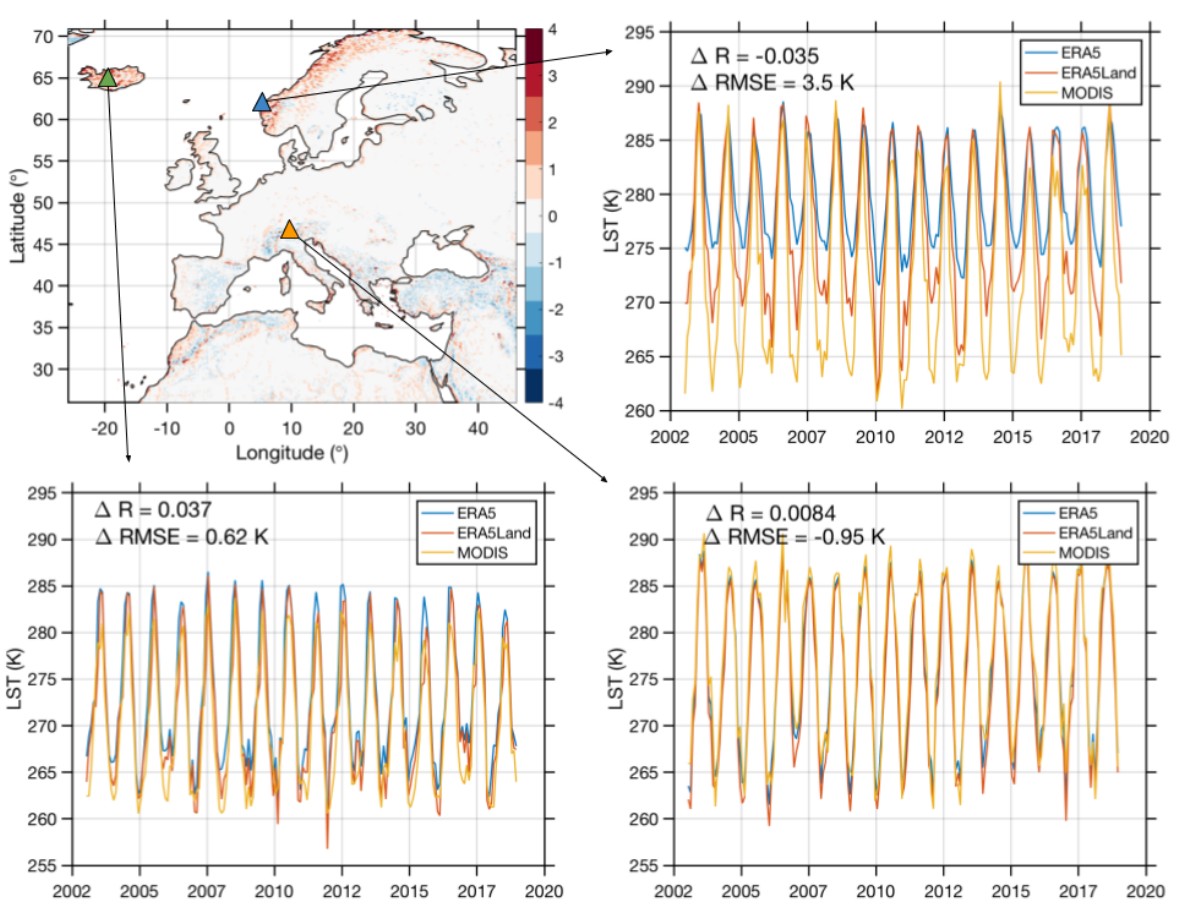

**Figure 20.** Map of Root Mean Square Error differences (ΔRMSE (K)) between ERA5 and ERA5-Land with respect to the MODIS LST average ensemble for the time period 2003-2018. Red values denote lower RMSE of ERA5-Land with respect to the observational record. Time series of ERA5, ERA5-Land and MODIS time series at selected pixels: coast of Norway (5.3 E, 61.6 N), the Alps (11.15 E, 47.4 N), and Iceland (20.26 W, 64.5 N). The quantitative differences in RMSE (ΔRMSE) and R (ΔR) between ERA5 and ERA5-Land with respect to MODIS LST at each location are reported.

## 5 Discussion and conclusions

This paper presents the new global ERA5-Land reanalysis. When the historical part is completed it will provide a detailed
record of the land surface evolution from 1950 to present through a series of key land surface variables representing the water and energy cycles. The temporal resolution is hourly and the horizontal resolution is 9 km, making it unique and suitable for a growing number of land surface applications. The quality of ERA5-Land fields was evaluated by direct comparison to a large number of *in situ* observations collected mainly for the period 2001-2018, as well as by comparison to additional model or satellite-based global reference datasets. The evolution of the operational ERA series of reanalyses was also assessed by





adding ERA-Interim and ERA5 global reanalyses to the evaluation exercise. Key components of the water and energy cycles
       were analyzed. Overall, the water cycle is improved according to the different variables evaluated, whereas the energy cycle
       variables evaluated in this paper show similar performance between ERA5 and ERA5-Land, and both performing clearly better
       than those provided by ERA-Interim:

       (a) Significant improved correlation of the ERA5-Land (compared to ERA-Interim and ERA5) root-zone soil moisture ab-
solute and anomaly values with respect to in situ measurements is achieved for almost all networks assessed. The root-zone is
       characterized by a slow temporal variation and thus a good initialization is crucial to reach equilibrium on relatively short time
       scales. ERA5 root-zone soil moisture is penalized by initialization from ERA-Interim, which leads to a multi-annual artificial
       trend of the time series over arid to semi-arid climates, where the standard deviation of soil moisture is low. On the contrary,
       the shallow surface layer responds quickly to short-term meteorological forcing variables, and hence, for the top layer, while
ERA5-Land only slightly improves on ERA5 estimates at the reference stations, improvements are not statistically significant.
       The main added value of high resolution in the top layer is to account for the correct soil type that changes the saturation level
       and the minimum level at which evapotranspiration no longer occurs. Consequently, ERA-Interim, with coarser resolution,
       shows the lowest performance of the three reanalyses datasets. Supporting these findings, ERA5-Land participated in a soil
       moisture intercomparison of 18 products against large number (826) of in situ stations and performed strongly (Beck et al.,
2021).

       (b) ERA5-Land snow mass and snow depth estimates are improved on mid-altitude mountains, where the higher resolution
       provides better orographic contours and the temperature correction is also important. However, ERA5 snow depth estimates
       match measurements slightly better at the highest mountains (>3300 m). At these altitudes, uncertainties in both the forcing
       and in the model parametrization are larger and can contribute to the growth of errors; processes like snow transport and subli-
mation, which are not considered in the current snow scheme formulation, can be important at these altitudes, as well as errors
       in the amount of solid precipitation. It is also worth noting that for high altitude stations, the spread of errors is larger in ERA5
       compared to ERA5-Land, which could lead to error compensation. The results per continent show the best performance of
       ERA5-Land snow fields in the US, especially in complex orographic areas, where spatial resolution is foremost. It implies a
       more accurate near surface air temperature, especially over mountains, which leads to a better representation of the sensible
heat flux and therefore better estimates of the snow depth. Also, there is a limited number of snow depth in situ observations
       assimilated in ERA5 over the US (de Rosnay et al., 2015), and therefore the assimilation system hardly compensates for sys-
       tematic model/snowfall biases. Contrarily, ERA5 performs better over Scandinavian countries. In general snowfall in the IFS
       suffers from a systematic positive bias (snow overestimation) in Europe. Although data assimilation is intended to correct for
       random errors, snow depth reports assimilated in ERA5 alleviate the systematic snow excess where these reports are assimi-
lated. The improvements of the snow scheme over the previous decade is also reflected in the overall better scores of ERA5
       compared to ERA-Interim.

       (c) LSWT is slightly improved in ERA5-Land at hourly and daily averaged steps with respect to in situ data in sites in Portugal
       and Finland. Note however that at punctual times of the warm period in Alqueva, sudden jumps of the ERA5-Land mixed
       layer depth leads to unrealistic LSWT. For instance, on 3 August 2018, from 8 am to 9 am local time, the mixed layer depth





jumped 5.3 m, leading to a 17.6°C increase and increasing the Alqueva reservoir LSWT to 40.7°C. Jumps are caused by a simplistic parametrization of the summer stratification in the lake scheme, as temperature is computed differently in calm and turbulent conditions. Ways to limit these jumps and to enable a smoother and more realistic transition are currently being investigated at ECMWF. The lake parametrization scheme relies heavily on accurate lake depth input data, and therefore poor data impacts its ability to accurately simulate lake parameters such as as LSWT. The higher resolution of ERA5-Land allows

for a more accurate specification of lake depth. Isolating lakes with more realistic depths in ERA5-Land reduces the LSWT errors by more than 20%. The positive influence of a higher resolution atmospheric forcing was also verified by isolating lakes which depth remained unchanged either using ERA5 or ERA5-Land. A comparison of LSWT estimates with a wider global database shows that, excluding exceptional lakes where Flake performs poorly, the performance of ERA5 and ERA5-Land LSWT is quite similar. The uncertainty of lake depths in the inventory used for validation is larger and could have an impact

on these results. But even by separating lakes with a depth less than 50 m (Flake was designed for medium-depth lakes) the performance of ERA5-Land LSWT does not improve compared to ERA5. This might be due to the averaging technique used for the in situ measurements, which uses satellite measurements representing one instant in time rather than continuous hourly data. In summer months, ERA5 LSWT biases are on average 2.2°C cooler than observations, whereas ERA5-Land LSWT is just 1.3°C cooler.

(d) River discharge integrates the various components of the water cycle. With improved soil moisture, snow and lakes in ERA5-Land, all these being physical inputs of river discharge generation, it is not surprising that the GloFAS river discharge obtained overall better scores when used with input data from ERA5-Land (GloFAS-ERA5-Land). This improvement in the skill is likely due to a number factors, such as higher horizontal resolution (from which smaller catchments benefit), improved representation of root-zone soil moisture and temperature (which contributes to increase the correlation and reduce biases with

river discharge observations) and snow processes (which should be an advantage in northern latitude catchments). Also, the fact that ERA5-Land does not directly assimilate any observation, as is the case for ERA5, has shown to have positive impacts on the closure of the water balance (Zsoter et al., 2019). The exception is in the West US and Amazonian basins. For the former, a later and larger snow melt due to an excess of accumulated snow over high peaks could lead to a decrease in correlation and increased biases with respect to in situ observations. The degradation in the Amazon is small, but it is linked to

an underestimation of the surface and subsurface runoff (not shown).

(e) ERA5-Land shows modest improvements in slhf and the Bowen ratio compared to ERA5 estimates, although the sshf is slightly better in ERA5, with the largest disagreements mainly found at high altitudes, where values are more extreme and errors are accumulated. The high resolution of ERA5-Land seems to be the reason for providing better fluxes at stations near the coasts or lakes. Under water stress conditions (e.g., central US, California and Australia) the slhf in ERA5-Land appears to

be more accurate than the ERA5 counterpart. In Europe the results are mixed. Nonetheless the fluxes in the alpine region are better in ERA5. This result coincides with lower performance of ERA5-Land (compared to ERA5) in terms of snow and river discharge estimates in this area when compared to in situ measurements. A possible cause may be the overestimation of snow depth in high altitude peaks in ERA5-Land (which does not benefit from the partial removal of snow by data assimilation), which in turn may lead to a late snow melt and vegetation exposure to the atmosphere which would influence transpiration.





Both ERA5 and ERA5-Land outperform GLEAM estimations of turbulent fluxes. The advantage of GLEAM over reanalysis lies in its simplicity and its ability to run on remote sensing forcing. However, through the use of common reanalysis forcing, the higher realism of the more complex model in ERA5–Land (CHTESSEL) is evident. The latter includes multiple processes that appear relevant for the surface energy partitioning, and are not explicitly represented in simpler models designed for the remote sensing of evaporation, such as GLEAM. All altogether, this paper did not provide clear evidence of an overall superior

performance of ERA5-Land surface energy fluxes over ERA5. While these results are inconclusive, one should bear in mind that they are only evaluated over 65 eddy-covariance sites. To better understand the possible added value of ERA5-Land turbulent fluxes, a more detailed evaluation based on towers located in contrasted climatic conditions and including other global reference datasets is recommended.

(f) The skin temperature (in this paper referred as LST), as a variable reacting quickly to any change in surface fluxes, shows

modest improvements in ERA5-Land compared to ERA5. However, as it has been the case for all variables evaluated in this paper, ERA5 and ERA5-Land LST are improved when compared to ERA-Interim. ERA5-Land obtains the lowest global averaged RMSE with respect to MODIS data, which is partly due to the contribution of the coastal points that simulate better the amplitude of the annual cycle of LST, and is a consequence of the higher spatial resolution of ERA5-Land. Other small disagreements between ERA5 and ERA5-Land are found over complex terrains, but they do not seem to favour any particular

reanalysis.

With the above results, one can conclude that the horizontal resolution matters and is a very important aspect in the accurate simulation of the spatial and temporal evolution of the hydrological cycle. However, this paper could only provide evidence of a modest improvement of the radiative aspect of ERA5-Land compared to ERA5. The latter conclusion is based on an

evaluation with respect to a small number of available samples. Other important aspects of the added value of ERA5-Land are the production speed (that allows cutting-edge land surface advances to be incorporated more rapidly) and the consistency presented over multi-decadal time scales, all of them making ERA5-Land a state-of-the-art dataset for multiple land applications. Finally, it is important to emphasize that an exhaustive evaluation of all land variables simulated in ERA5-Land is not feasible in a single paper. While this paper provides significant validation elements to demonstrate the added value of ERA5-Land, the

wider scientific community is invited to carry out more detailed evaluation of individual components. For example, a more extensive validation of soil moisture following international agreed best practice (Gruber et al., 2020) is highlighted as an area for further research. With the public release of ERA5-Land, research and development has already shown some caveats of the dataset, such as the treatment of soil temperature in permafrost regions (Cao et al., 2020). Further studies comparing with alternative well-referenced sites, as well as with regional and global datasets is encouraged. For instance Pelosi et al. (2020)

compared UERRA regional reanalysis (Copernicus Climate Change Service, 2020) forced by ERA-Interim and ERA5-Land to assess the performance of evapotranspiration estimates based on weather data in the South of Italy.





## 6   Perspectives

The ERA5-Land dataset presented in this paper is the first operational land reanalysis of the ERA series. This paper has demonstrated its added value by comparing ERA5-Land estimates to a wide range of in situ observations, ERA-Interim and ERA5
reanalyses. Despite the overall observed improvement of the land states in ERA5-Land, there is scope for improvement for several components, which will be the focus for the construction of future enhanced versions. They are discussed below;

In the context of C3S, all operational products require an estimate of the associated uncertainty. Currently, estimates of uncertainty of ERA5-land variables are those corresponding to the ERA5 counterpart. ERA5 uncertainties are obtained by
running a 10-member EDA, which also provides the background-error estimates for the deterministic HRES 4D-VAR data assimilation system (Hersbach et al., 2020). First tests conducted at ECMWF running an ensemble of offline simulations with an ensemble of initial conditions and atmospheric forcing provided by the 10-member EDA of ERA5, indicate that the spread of the land surface variables is unrealistically low (not shown). The likely reason is that in the production of the ERA5 ensemble the physics of the surface model is not perturbed, i.e., the surface model is assumed to be perfect. To obtain realistic
uncertainties, not only input and forcing parameters to the land surface model should be perturbed, but also key variables and parameters of the surface scheme (e.g. MacLeod et al., 2016; Orth et al., 2016), adding the contribution of land surface model error to the ensemble spread. Future studies are envisaged to investigate this path to provide meaningful uncertainties to land reanalysis.

One of the most important driving variables of ERA5-Land is precipitation. Systematic model-based precipitation biases can
potentially spread into the land state estimates. ERA5 precipitation (used as input forcing of ERA5-land) benefits from a much improved data assimilation system assimilating millions of extra observations compared to its predecessor ERA-Interim. Indirectly, ERA5-Land benefits from these extra observations as well. However, ERA5 still has large precipitation biases, especially in tropical regions. A recent bias-adjusted dataset based on ERA5 has been produced for impact studies with a 0.5° resolution (WFDE5, Cucchi et al., 2020). That study demonstrated the added value of the bias corrections on large-scale hydrological
modeling. Future versions of ERA5-Land could consider similar approaches taking into account such bias-corrected coarser resolution forcing datasets as well as their availability in near-real-time.

The land surface model used in ERA5-Land, CHTESSEL, also has the option to estimate carbon fluxes and its coupling with plants transpiration through the A-gs formulation (Jacobs et al., 1996 ; Calvet et al., 1998; Boussetta et al., 2013a). This module is operationally active for the vegetation and it allows estimates for the carbon fluxes in a modular approach with evaporation
being computed through a resistance approach (Jarvis et al., 1976). This choice is adopted since its integration in the operational numerical weather prediction still requires further developments (Boussetta et al., 2013a). Although carbon fluxes are an actual output of ERA5-Land, they are not made available because of persistent biases. Agustí-Panareda et al. (2016) implemented a Biogenic Flux Adjustment Scheme (BFAS), which uses the Copernicus Atmosphere Monitoring Service (CAMS) inversion by Chevallier et al. (2010) to correct for the biases in the 10-day budget of the modelled carbon fluxes at continental scales. This
is currently being extended for usage in the FLUXCOM product (Jung et al., 2020) in order to bias correct the two components



of the biogenic fluxes (Gross Primary Production and ecosystem respiration) separately. Figure 21 shows carbon fluxes from the ERA5-Land simulation with and without the BFAS bias correction compared to observations from the ICOS-ETC network (ICOS-ETC Drought 2018 Team, 2019). The impact of the bias correction is largest during boreal summer when the vegetation is most active and the soil respiration is also largest. A future version of land reanalysis may include the bias-corrected carbon

fluxes which will also benefit from the ongoing and future land surface model developments associated with the representation of the vegetation.

Another important research path with large potential is the revision and use of dynamic auxiliary data. Currently, ERA5-Land assumes a fixed land cover, whereas Leaf Area Index (LAI) and albedo are based on a static monthly climatology. The former assumes that land cover remains unaltered for the complete reanalysis period and that cities are non existent, whereas

the latter will not be able to accurately represent more frequent LAI anomalies. Based on the research conducted in the ESA-CCI programme, C3S provides (through the Climate Data Store (CDS)) climate data records of land cover at yearly frequency, as well as a close to real time global maps of LAI with 10 to 20 days latency. Moreover, recent studies have identified errors in the diurnal cycle of land surface temperature over Iberian Peninsula associated with the current land cover used in CHTESSEL and 1 hour model time steps (Johannsen et al., 2019; Nogueira et al., 2020). Therefore, the revision of the land cover and LAI to

a new database as well as the introduction of their inter-annual variability is expected to provide more realistic land conditions as input to the surface scheme. There are also ongoing efforts to revise the vertical discretisation of the vegetation roots and soil layers distribution (e.g. Mueller-Quintino et al., 2016; Stevens et al., 2020), which could complement the land cover and vegetation updates.

Finally, the coupling of the offline simulations to an offline data assimilation system is a promising approach. The advantages

are multiple, in particular the assimilation of local datasets (precipitation radar observations, local soil moisture networks, etc.), permitting small adjustments of land state estimates. Undoubtedly, this will go accompanied by an increase in computational cost. Ongoing efforts to improve the parallelization of the land model will allow global high resolution simulations for several decades to be performed at affordable computational costs.

All the components discussed above are currently under research and, along with the improvement in other components of the land surface scheme (increase of the number of soil layers, introduction of a multi-layer snow scheme, reduction of model time step, etc.), provide the basis for a future new version with improved accuracy for the land states at multi-decadal time scales.

## 7    Data availability

ERA5-Land data are available through the C3S CDS, and at the time of writing this paper the data are available from January 1981. The data are accessible either via the user interface (https://cds.climate.copernicus.eu/cdsapp#!/dataset/reanalysis-era5-land?tab=overview) or through the CDS Application Program Interface (API). The data are updated with 2-3 months delay with respect to real time. However, a close to real time facility is planned to be implemented in 2021. The atmospheric forcing used

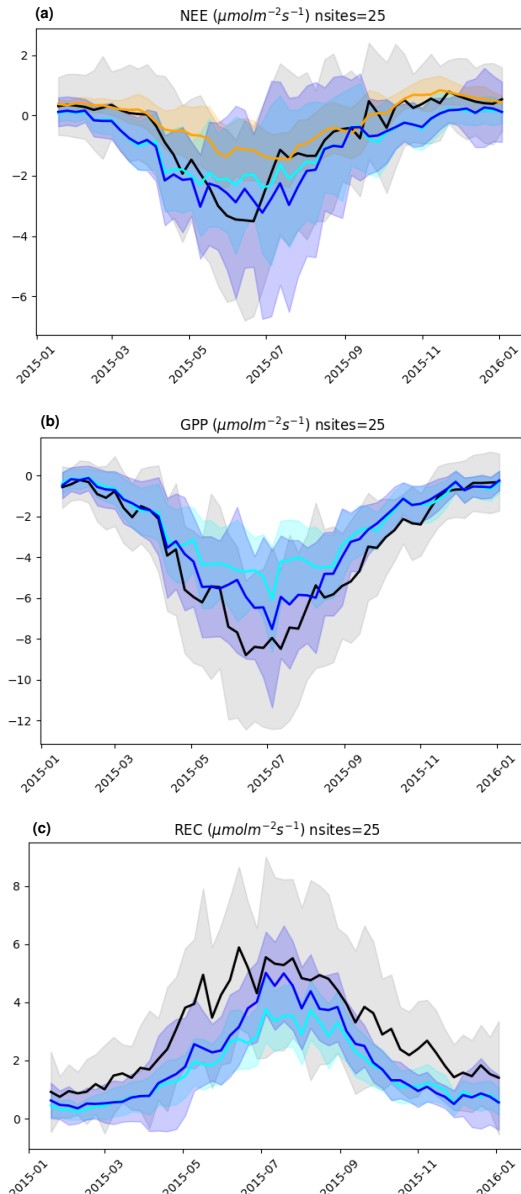

**Figure 21.** Mean seasonal cycle of weekly $CO_2$ biogenic fluxes $\mu$mol [m$^{-2}$ s$^{-1}$] from ERA5-Land (in cyan), bias-corrected ERA5-Land (in blue), and the CAMS atmospheric in situ inversion product (v18r3, https://atmosphere.copernicus.eu/data, (Chevallier et al., 2010)) minus fossil fuel emissions (in orange) for 2015 at 25 ICOS-ETC sites over Europe (in black) (ICOS-ETC Drought 2018 Team, 2019). The shading shows the standard deviation of each data set across the 25 sites. Top panel: Net Ecosystem Exchange (NEE); middle panel: Gross Primary Production (GPP); lower panel: Ecosystem Respiration (REC).





to drive the land simulations is also available and already interpolated to the ERA5-Land grid. Note that in the CDS, and for

user convenience, the data have been interpolated to a regular lat-lon grid of 0.1°resolution.

All ERA5-land fields are made available at hourly temporal resolution and post-processed as monthly means. Monthly means are easier to handle and faster to retrieve, which is especially well suited for climate studies and also addresses an important requirement of reanalysis users. Two types of monthly means are post-processed:

- monthly means of daily means (average over all the hourly fields in a month)

$$<\mathbf{x}>= \frac{1}{N_d}\frac{1}{N_s}\sum_{d=1}^{N_d}\sum_{s=1}^{24}\mathcal{M}(\mathbf{x})_{d,s} \quad (6)$$

- monthly means of synoptic means (averaged for a specific time of the day in a month)

$$<\mathbf{x_s}>= \frac{1}{N_d}\sum_{d=1}^{N_d}\mathcal{M}(\mathbf{x})_{d,s} \quad (7)$$

with $< \mathbf{x} >$ the estimate of the monthly average for the field $\mathbf{x}$, $N_d$ the number of days in a month, $N_s$ the number of forecast steps in a day, $\mathcal{M}$ the forecast model, $d$ the day and $s$ the forecast step from 00UTC in a 24 h cycle. Note that the water and

energy fluxes are accumulated from the beginning of the forecast time at 00UTC with a maximum of 24 h accumulation period. For example, the monthly mean of surface runoff at 12UTC will provide the monthly averaged runoff accumulated from 00 to 12UTC. Therefore, for water and energy fluxes, Eq. 6 becomes:

- monthly means of daily means for water and energy fluxes

$$<\mathbf{x}>= \frac{1}{N_d}\sum_{d=1}^{N_d}\mathcal{M}(\mathbf{x})_{d,s=24} \quad (8)$$

Further technical details are also provided in the online documentation (https://confluence.ecmwf.int/display/CKB/ERA5-Land% 3A+data+documentation).

ERA-Interim surface data used in this study is freely available through the ECMWF catalogue: https://apps.ecmwf.int/datasets/ data/interim-full-daily/levtype=sfc/

ERA5 hourly data on single levels used in this study is also freely available through the C3S CDS (doi: 10.24381/cds.adbb2d47).

All soil moisture data used for validation is available through the International Soil Moisture Network, https://ismn.geo. tuwien.ac.at/en/

Access to the lake data used for evaluation in this paper is accessible as follows:

- The Alqueva reservoir hourly data were provided by the Portuguese University of Evora, data are provided by demand, personal communication with Miguel Potes, Maksim Iakunin and Rui Salgado;



- 27 Finnish lakes daily data provided by Finish Environment Institute SYKE are open access and available at http://rajapinnat.ymparisto.fi/api/Hydrologiarajapinta/1.0/;

- Summer months (June, July and August) averaged values from the global inventory "Globally distributed lake surface water temperatures collected in situ and by satellites; 1985-2009" are freely available at https://portal.lternet.edu/nis/mapbrowse?packageid=knb-lter-ntl.10001.3.

The ESM-SnowMIP dataset used for the evaluation of the snow fields is available at https://doi.pangaea.de/10.1594/PANGAEA.897575 (see also Ménard et al., 2019). The snow depth dataset from the GHCN-daily network is available at https://www.ncdc.
noaa.gov/ghcnd-data-access; the version used is v3.24 (Menne et al., 2012b).

The majority (i.e. 75 %) of river discharge observation stations used in the evaluation are openly available from the Global Runoff Data Centre (GRDC): https://www.bafg.de/GRDC/EN/Home/homepage_node.html. The remaining stations have been shared by GloFAS partners worldwide to improve spatial coverage. The benchmark river discharge reanalysis
dataset, GloFAS-ERA5 (version 2.1), is openly available from the CDS: https://cds.climate.copernicus.eu/cdsapp#!/dataset/cems-glofas-historical?tab=overview with doi: 10.24381/cds.a4fdd6b9.

GLEAM data were accessed from https://www.gleam.eu/ and the FLUXNET2015 Tier2 data set can be accessed from the FLUXNET data portal at https://fluxnet.fluxdata.org/data/fluxnet2015-dataset/. This work used eddy-covariance data ac-
quired and shared by the FLUXNET community, including these networks: AmeriFlux, AfriFlux, AsiaFlux, CarboAfrica, CarboEuropeIP, CarboItaly, CarboMont, ChinaFlux, Fluxnet-Canada, GreenGrass, ICOS, KoFlux, LBA, NECC, OzFlux-TERN, TCOS-Siberia, and USCCC. The FLUXNET eddy covariance data processing and harmonization was carried out by the ICOS Ecosystem Thematic Center, AmeriFlux Management Project and Fluxdata project of FLUXNET, with the support of CDIAC, and the OzFlux, ChinaFlux and AsiaFlux offices.

MODIS LST data from Aqua (doi: 10.5067/MODIS/MYD11C3.006) and Terra (doi: 10.5067/MODIS/MOD11C3.006) platforms are available through the Land Processes Distributed Active Archive Center (https://lpdaac.usgs.gov/).

FLUXNET-2015 data used for the evaluation of GPP, NEE and REC is available from the Drought-2018 ecosystem eddy
covariance flux product in FLUXNET-Archive format - release 2019-1 (Version 1.0). ICOS Carbon Portal, //doi.org/10.18160/PZDK-EF78.

*Author contributions.* JMS produced the ERA5-Land dataset, designed the study and wrote the paper. ED, GB, SB, HH, DM, CB and JNT revised the paper and the figures, wrote some text and provided a critical review of the paper, CA pre-processed soil moisture data and NJRF conducted the corresponding analysis, GA processed and analysed the snow data, MC collected, processed and analysed all the lake data of
this study, SH and EZ provided the evaluation of the river discharge data, BM processed and evaluated all the surface energy fluxes and run





the GLEAM model, MP collected and evaluated the skin temperature data, AAP processed and produced the bias-corrected carbon fluxes example. SH also proofread most of the paper.

*Competing interests.* To our knowledge there is no conflict of interest of the material presented in this paper.

*Acknowledgements.* ERA5-Land was produced with funding from the European Union's Copernicus Climate Change Service. GloFAS partners and the Global Runoff Data Centre (GRDC), 56068 Koblenz, Germany, are thanked for providing river discharge observations. Laura Rontu and Matti Horttanainen are thanked for providing in situ data for Finish lakes. Miguel Potes, Rui Salgado and Maksim Iakunin are thanked for providing in situ data for Alqueva reservoir. Thanks to Alex Vermeulen (ICOS Carbon Portal) for his advice on the use of the Drought-2018 ecosystem eddy covariance flux product in FLUXNET-Archive format - release 2019-1. The contribution of GA was funded
through the APPLICATE project. The APPLICATE has received funding from the European Union's Horizon 2020 research and innovation programme under grant agreement 727862.



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
