# Peer review of "ERA5-Land: A state-of-the-art global reanalysis dataset for land applications"

_Earth System Science Data, 2021_

## Referee Comment (RC2)

[referee-annotated manuscript omitted]

---

## Author Response (AR1)

**Responses to reviewers:**

**Reviewer-1:**

1) **In line 6 of the abstract you mention that ERA5-Land enables the characterisation of trends. Yet, the manuscript does not include any analysis of trends. Why were trends not assessed, given that likely the product will be intensively used for trend analyses?**

Thanks for your comment. We don't intend to give the impression that this paper also includes analysis trends. The latter is out of the scope of the paper and we believe that trends' analyses deserve a dedicated study, whereas this paper is more focused on the presentation of the ERA5-Land dataset.

Therefore, in order to make it clearer we have replaced in the abstract:

"ERA5-Land describes the evolution of the water and energy cycles over land in a consistent manner over the production period, enabling the characterisation of trends and anomalies" by

"ERA5-Land describes the evolution of the water and energy cycles over land in a consistent manner over the production period, which among others could be used to analyse trends and anomalies".

Also, to align with the comments of the second reviewer, we also have added a sentence in the "discussion and conclusions" section in lines 663-665 of the revised manuscript:

"The reduced impact of discontinuities by using longer spin-up periods, could also be a crucial factor to obtain accurate trends over multi-decadal periods for variables slowly changing in time, for instance the root-zone soil moisture (see bottom panel of Fig.3)".

2) **The benefits of ERA5-Land wrt ERA5 are not ubiquitous. Clearly, the resolution is higher, but this doesn't always lead to better skill. To convince the user of using a ~10fold larger dataset and exponentially increasing subsequent computation times for further use, could you summarise at the end of the manuscript where, for which variables, and for what applications ERA5-Land shall be favoured over ERA5?**

Thanks for your comment. As a matter of fact, ERA5-Land is in size much smaller than ERA5 (TB vs PB) since is limited to the land surface layer. However, it is still true that for those fields which are common to ERA5 and ERA5-Land, the latter will be, due to the higher

resolution, larger in volume, despite that the data over oceans in ERA5-Land has been masked out and therefore does not inflate the total volume of the field.

The main objective of this paper is the presentation of ERA5-Land while providing the reader sufficient evaluation elements (through the evaluation of several key variables of the water and energy cycles) to guarantee the good quality of the ERA5-Land dataset. At this respect and in wider terms, due to its specific characteristics (hourly frequency, multi-decadal, consistency, higher resolution) we believe ERA5-Land is ideal for all type of land applications, but the choice of an ERA5 or ERA5-Land common field will depend very much on the specific application. Relevant questions are: are data volume and data handling limiting factors? Are resolution and accuracy the most crucial factors? Is the analysis applied at global, regional or local scale? What is the computational power for the required application? In the 'conclusions and discussion' section, we have provided specific comments for each of the variables evaluated in this paper (and to make it clearer which variables are we discussing/concluding, we have replaced the a), b), c)… by the relevant variable name), but it is however impossible to produce a recommendation list for each application. Following the reviewer comment, we have added the following sentence to last paragraph of the 'conclusions and discussion' section, Lines 670-672 of the revised manuscript:

"While, in wider terms, we recommend the use of ERA5-Land fields over ERA5 for all type of land applications, one should factor in their choice elements such as available computer and data handling resources, importance of spatial resolution versus data volume, area of application, temporal consistency, etc."

**3) I appreciate that the new product can be comprehensively assessed for all fields, but can you at least justify why these fields were evaluated and others not?**

We assume the reviewer means "cannot be comprehensively assessed for all fields". Our choice was based on four factors: a) key variables of the water cycle, b) key variables of the energy cycle, c) availability and accessibility of evaluation data at the time of the study, d) inclusion of global scale evaluation by using reference satellite datasets. Based on all the four previous factors, we came up with the set of selected variables and reference datasets. We agree that the list of variables and reference datasets selected are not exhaustive, for instance 2 m temperature evaluation and soil temperature data from the ISMN as reference dataset could also have been added to this study. However, the length of the paper was a constrain and we had to prioritise some choices (see also our response to the next point). Our hope is that indeed, as stated in lines 647-651 of the original manuscript, the wider scientific community engages in more in-depth analyses of other variables, as it has been the case with the ERA5 atmospheric reanalysis.

To make this point more clear, we have additionally added the following sentence to lines 183-184 of the revised manuscript:

"Note that the list of evaluated variables and reference datasets is not exhaustive and was based on factors such as availability of data at the time of the evaluation".

4) **Similarly, one needs to make choices regarding the reference data to be used. The choice seems slightly arbitrary and hence, the conclusions drawn can be inconsistent between fields. Especially, as ERA5-Land is embedded in the C3S Climate Data Store I would have expected that the assessment would draw on the various (satellite) products provided by this service. Vice versa, for LST you used a satellite product but you could also have used soil temperature data from the ISMN.**

Although ERA5-Land is available from the C3S Climate Data Store, the evaluation exercise started much before that some of the current satellite datasets in the CDS were available. Evaluation against satellite data is not trivial, and requires dedicated efforts for each dataset considered. Our primary choice has been using in-situ ground data as 'ground truth', but we didn't want to limit ourselves to in situ only.

For skin temperature, in situ soil temperature measurements from ISMN are representative of soil at different temperatures, while we aimed at validating the joint contribution of soil (top surface) and vegetation into skin temperature, which is more closely related to satellite estimates from thermal sensors. Among the different sensors, we chose MODIS for its spatio-temporal coverage, since it provides four daily global long-term observations combining Terra and Aqua platforms, and daytime and night time overpasses. Note this MODIS ensemble has in turn been validated with in-situ data from tower-based measurements (see Chen et al., 2017).

Since, in this paper, the evaluation exercise cannot be exhaustive for all fields, we have been guided by the aim of being representative and relevant in our choices for selected and independent validation results, while keeping the size of the paper limited. Please, see also the response to the previous comment.

Minor issues

- **Line 227: please also cite the recently published paper on the ISMN: https://doi.org/10.5194/hess-2021-2**

  Done (now in line 232 of the revised paper)

- **Did you also apply the quality flags of the ISMN data? https://doi.org/10.2136/vzj2012.0097**

  No, although we acknowledge the usefulness of such a flags, the quality flags of the ISMN data were not used in this study.

- **Line 391: From the boxplots one cannot tell that ERA5-Land on average performs slightly better than ERA5, as the results for the different regions are based on a different number of stations. Does this statement still hold if you perform the analysis for all stations in these regions together?**

  We have produced the boxplots adding all the stations used of the ISMN, independently of the region. The plots are generated for the top soil layer, as for the root-zone we only had observations for North-America. As the reviewer can see, the boxplots are very similar to those shown in Fig. 7 (a-d), as most of the stations are located in North-America. The conclusions are the same, so we prefer to keep the current boxplots to show the break-down per regions, and which also show quite clearly that the impact of ERA5-Land is larger in the root-zone.

[Figure]

*Figure 1.- Box plots showing the evaluation metrics of ERA5-Land and ERA5 against in situ measurements of the ISMN network at 5 cm. On each box, the central mark indicates the median, and the bottom and top edges of the box indicate the 25th (q25) and 75th (q75) percentiles, respectively. The whiskers extend to the most extreme data points not considered outliers.*

[Figure]

*Figure 2.- Box plots showing the evaluation metrics of ERA5-Land and ERA5 anomalies against in situ measurements of the ISMN network at 5 cm. On each box, the central mark indicates the median, and the bottom and top edges of the box indicate the 25th (q25) and 75th (q75) percentiles, respectively. The whiskers extend to the most extreme data points not considered outliers.*

Following the reviewer comment, we have added in the results section a paragraph (Lines 418-424 of the revised manuscript) to stress the fact that stations have also been compared globally: "The results were also analyzed site per site taking into account the confidence interval of the Pearson correlation obtained for ERA5 and ERA5-Land versus the in situ measurements. A correlation difference is considered significant if the confidence intervals do not overlap. The Pearson correlation difference for ERA5 and ERA5Land at 5 cm is significant for 382 sensors, of which 64 % show higher correlation for ERA5 Land. The correlation difference at 20 cm is significant for 417 sites, of which 72% show a higher correlation for ERA5 Land. Finally, the correlation difference at 50 cm is significant for 479 sites, of which 88% show a higher correlation for ERA5 Land".

- **URL of the ISMN: change into ismn.earth**

  Thanks to point this out. Done.

- **As this is a dataset paper, it would be useful to include a list and short description of the input and output fields**

Following the reviewer recommendation, we have included an appendix to the manuscript with the list of input/output fields, and provided a reference for the description of each field in the caption of the table. The tables are also referenced in the main text, in lines 87 and 126-127 of the revised manuscript.

- **For ET: briefly describe the evaluation metrics, even as the fields sshf, slhf ( l492), and bowen ratio**

Thanks for this remark. We have removed the description of the evaluation metrics from the beginning of section 4.5.1, and moved them and slightly modified to the end of section 3.7.1. (lines 361-363 of the revised manuscript), as this is more consistent with the evaluation metrics described for each subsection.

We are not sure what the reviewer means by "even as the fields sshf, slhf (l492) and bowen ratio". If the reviewer means what these abbreviations stands for, they were defined in lines 334 and 335 of the original manuscript, section 3.7.1. However, in the revised version of the paper and also in line with the comments of reviewer-2, we have replaced all "sshf and slhf" entries by "H" and "λρE", respectively.

The paper provides a description of the ERA5-Land product and an evaluation of it mainly against in-situ but also one satellite-based dataset (skin temperature). The same evaluation is also done for ERA-Interim and ERA5, which indicates the added value but also limitations of ERA5-Land. Further, the downscaled ERA5-Land meteorological data (surface air temperature and surface net radiation) is used to force the land evaporation model GLEAM.

This paper is highly important. ERA5-Land will be to a great extent used, and this paper will be the first reference and primary inspiration and guidance to many. Overall, the paper is great, especially those parts in which performance differences between ERA5-Land and ERA5 are discussed are very interesting. However, I also see several aspects that need clarification and improvement. In general, figures/caption and, in some parts, also language should be substantially polished. I suggest a moderate revision. I provide a number of suggestions below and in the attached commented paper pdf.

(1) **I see mainly four reasons why ERA5-Land is supposed to be superior to ERA5:**

**(i) higher spatial resolution of static land model parameters**

**(ii) higher spatial resolution of downscaled forcing**

**(iii) reduced impact of discontinuities by longer spin-ups for the segments**

**(iv) newer LSM version used in ERA5-Land**

**How does trend analysis benefit from these four reasons? Indeed, ERA5-Land will be used for trend analysis but since it is not included in the evaluation, what are the expected improvements relative to ERA5? The skill improvement that is currently presented was probably mainly due to the higher spatial resolution, I am wondering how relevant (for which variables) higher spatial resolution is for trend analysis? If it is decided not to provide a trend analysis in this paper, it would be useful to read about the authors' opinion as a first guidance for the user community regarding trend analysis. Thinking of trend analysis of river discharge, it should be emphasized that crucial factors like land use change are not included in the land surface model. I recommend a discussion section on implications of the current results + reanalysis setups for trend analysis.**

Many thanks for your very interesting comments. We believe the comments about the trends would be very relevant if the paper would conduct a trend analysis. However, although we believe that this dataset enables trend analysis, it is out of the scope of this paper. Trend analysis deserves a dedicated study and we cannot fit it in a data presentation

paper which is already quite long by itself. We have tried to make it clear in Lines 5-6 of the abstract (see response to comment (1) of reviewer-1), and also we have added a sentence in line 662 of the "conclusions and discussion" section of the revised manuscript:

Line 662 "[...] and the consistency presented over multi-decadal time scales (that could set the basis to enable reliable trend analyses), all of them making ERA5-Land a state-of-the-art dataset for multiple land applications".

It is difficult to provide an educated guess of how ERA5-Land trends will compare to those of ERA5 without conducting the analysis itself. This paper has shown that, based on the evaluation carried out over several key variables of the water cycle, ERA5-Land is more accurate at least at representing the spatial/temporal evolution of the hydrological cycle. The differences with ERA5 are not enormous though, so we believe that trends will not be very different at global scale. However locally, where resolution matters, it could provide more accurate trends. In particular trends at areas with low soil moisture variability, where the reduced impact of discontinuities by longer spin-ups is important, could benefit from ERA5-Land multi-decadal consistency. At this respect we have also added a sentence in the discussion part, in lines 658-660 of the revised manuscript:

"The reduced impact of discontinuities by using longer spin-up periods, could also be a crucial factor to obtain accurate trends over multi-decadal periods for variables slowly changing in time, for instance the root-zone soil moisture (see bottom panel of Fig.3)".

We agree that factors such as land use change are crucial for accurate land surface modelling and trend analysis, and those are in the current plans of the ECMWF land surface model evolution. This point, as well as the use of a monthly (static) LAI, has already been described in the 'perspectives' section, in lines 697-708 of the original manuscript.

(2) **One application of ERA5-Land will be to force other land models with higher temporal and spatial resolution atmospheric input. An application is indicated by the evaluation with GLEAM, though differences between ERA5 and ERA5-Land driving GLEAM were very small. Consider to extend the related discussion section (e) and include a discussion of the benefits from downscaled precipitation for other land models.**

This is a really good point raised by the reviewer. A short discussion on precipitation is already included in the paper in the "perspectives" section, however we have also extended it as to comment on the benefits from downscaled precipitation. In the perspectives, after "as well as their availability in near-real-time" In lines 702-704, we have added the following paragraph:

 "In addition to coarse-scale near-real-time bias-correction, high resolution downscaling could be also explored. Such a correction could be based on a climatological rescaling of precipitation based on a high resolution reference climatology (e.g. Karger et al. 2017)"

For Karger et al. 2017, the new reference is:

Karger, D. N., Conrad, O., Böhner, J., Kawohl, T., Kreft, H., Soria-Auza, R. W., Zimmermann, N. E., Linder, H. P., and Kessler, M.: Climatologies at high resolution for the earth's land surface areas, Sci. Data, 5, 170 122, https://doi.org/10.1038/sdata.2017.122, 2017.

(3) **The new PET variable is included in the portfolio of ERA5-Land, which is briefly mentioned with a note of caution on its use because the atmosphere is not affected by the water-unlimited land surface assumed for PET. That seems to me being a general issue of PET. How is this different for the ERA5-Land PET from other PET approaches e.g. discussed in https://doi.org/10.5194/hess-23-925-2019 ?**

We agree with the reviewer on this comment and we acknowledge that the main problem of the PET concept is that it is related to a hypothetical quantity defined for hypothetical environmental conditions which hampers experimental validations. The ERA5-Land PET is only one option among others such the Priesley-Taylor formulation or the radiation-based ones. A rational of the concept of evaporation under unstressed soil moisture conditions would require mapping effects on net radiation (which is accounted for in the second call to the surface model) without atmospheric feedbacks. However, there is always a hypothetical element in the atmospheric feedback assessment, as it depends on the spatial extent of the target unstressed area. Small plots in arid environments will generate advection effects, but large areas will lead to atmospheric moistening considerably supressing evaporation, and advection effects confined to the edges of the considered area. This to emphasize the complexity of the PET estimation problem and that it would depend on the target application being dependent on the spatial (and temporal) scale and eventually on the irrigation regime.

Also to note that PET can also be estimated at a daily time scale with the Priestly-Taylor formulation using daily averages of net radiation, temperature, humidity, and wind from ERA5/ERA5-Land without need of model timestep integration (which is another option to the ERA users).

(4) **Results indicate higher bias for ERA5-Land energy fluxes. Was the EC data corrected for energy balance closure gap?**

The EC data was not corrected for energy balance closure. Eddy covariance In-situ flux measurements can be affected by systematic biases due to issues affecting the energy closure. However, the correction of those imbalances require making assumptions on the sources of the errors, which can also impact the fluxes. Furthermore all the measured fluxes (including ground-heat flux) are needed, and they are not available from many sites. Therefore, no correction of the EC data was considered in this study.

See an extended discussion of this point here below, in response to a comment on Line 340 of the original manuscript.

**(5) I found it interesting that the river discharge skill mostly improved for ERA5-Land relative to ERA5. Based on the snow depth evaluation, I would have assumed that discharge for watersheds with significant snow melt might deteriorate due to lacking assimilation of snow obs in ERA5-Land compared to ERA5. It could be discussed more in detail why snow DA did not help ERA5 to perform better. Further, river discharge of large-scale routing models are often evaluated not at daily but at coarser resolution, e.g. monthly, due to large uncertainties of routing parameters that makes it unrealistic to predict the timing of peaks at daily resolution. To be more comparable with other routing scheme results, Figure 13 (with reduced (a) figure, see annotated pdf) could be also provided for monthly resolution.**

As described in Zsoter et al. 2019, the snow data assimilation was shown to be detrimental in much of the Northern Hemispheric snow impacted areas in the ERA5 experiment, compared with the offline simulation without coupling and land data assimilation. The average snow increments in ERA5 are negative, due mainly to the too slow snow melt in the HTESSEL, which consequently  removes water  from  the hydrological system. This contributes  to  the decrease  of  the  dominantly  negative  biases in a large  area with generally deteriorating the hydrological performance.

To make this point more clear, we have added the following paragraph, in lines 621-627 of the revised manuscript, in the relevant part of the discussion section:

"Also, the fact that ERA5-Land does not directly assimilate any observation, as is the case for ERA5, has shown to have positive impacts on the closure of the water balance (Zsoter et al., 2019). They have shown that the  snow  data  assimilation is detrimental to the hydrology in large  parts of  the snow  impacted Northern  Hemisphere in  the  ERA5 experiment, compared with the offline simulation without coupling and land data assimilation (such as ERA5-Land). The average snow increments in ERA5 are negative, due mainly to the too slow snow melt in CHTESSEL, which consequently removes water from the hydrological system. This contributes to decreasing the dominantly negative biases in a large area with subsequently deteriorating the hydrological performance. The exception is in the West US and Amazonian basins."

With respect to the use of daily, rather than monthly; to be able to appropriately evaluate the  ability  of  a  model  to  simulate  river  discharge,  especially  with regards  to a model's ability to capture the timing of river discharge peaks, it is necessary to use a minimum of a daily model time-step, which is also the time resolution of the available observations. The  aim  of  the  evaluation  here  was  to  assess  the  ability of GloFAS-ERA5 compared to GloFAS -ERA5-Land in capturing key hydrological dynamics of timing, bias, and variability – performed with the modified Kling-Gupta Efficiency (KGE) statistical metric. This is a widely used standard methodology in hydrological model evaluation. We hence  prefer  to  maintain  the  analysis  of  river  discharge  using  daily  time-steps.

(6) **The main technical aspect of the production of ERA5-Land, that was not sufficiently explained to me, is how the "Integration of the land surface model in 24 cycles" works. Please provide longer explanation of what is done here and why.**

The "24-h cycles" is a convention that has been used to make it easier the retrieval of 'forecasted' fields by ERA5-Land users (as it was done for ERA-Interim) and limit the accumulation period of surface fluxes to 24h. ERA5-Land is run as a continuous simulation (in 24 hour's chunks with reproducible restarts) forced by ERA5 near-surface meteorological variables, so although theoretically there aren't 'analysis' fields, some of them are grib-coded as 'analysis' and others as 'forecasts'. In the C3S catalogue the 'forecast' fields are valid at a specific time of the day (00 to 23UTC). We could have equally selected 12h, 48h cycles or a different one, but it would have been much more complex to understand the validity time of a 'forecast' field.

We believe that the reason above is too technical to be explained in detail in a data presentation paper. In section 7 we have explained how the accumulation periods works, and we have directed interested readers about the technical details to the online documentation that provides further details.

(7) **Regarding water management applications, it's important to mention to the user community that ERA5-Land lacks a groundwater storage. Something that should be clarified at some place.**

The reviewer is right. We are currently working with a modular, flexible system called ECLand. This system allows to develop several modelling aspects separately, such as the open water areas and river/inundation. Currently there are some ongoing developments with coupling runoff with a river discharge system (CamaFlood), and indeed in the longer term the introduction of a groundwater storage will also be considered as part of our developments. We have added this point in the perspectives section, by replacing:

"All the components discussed above are currently under research and, along with the improvement in other components of the land surface scheme (increase of the number of soil layers, introduction of a multi-layer snow scheme, reduction of model time step, etc.), provide the basis for a future new version with improved accuracy for the land states at multi-decadal timescales" by

"All the components discussed above are currently under research, but they are not the only ones. ECMWF is currently working with a flexible, modular system called ECLand (Boussetta et al., 2021), which allows to develop separately several modelling aspects of the land surface, such as the increase of the number of soil layers or the introduction of a multi-layer snow scheme, as well as progressing on other longer term perspectives such as the introduction of a groundwater storage or the reduction of the model time step. All the above ongoing developments will provide the basis for a future new version with improved accuracy for the land states at multi-decadal timescales.".

The added reference is the following:

Boussetta, S., Balsamo, G., Arduini, G., Dutra, E., McNorton, J., Choulga, M., Agustí-Panareda, A., Beljaars, A., Wedi, N., Muñoz Sabater, J., de Rosnay, P., Sandu, I., Hadade, I., Carver, G., Mazzetti, C., Prudhomme, C., Yamazaki, D., and Zsoter, E.: ECLand: The ECMWFLand Surface Modelling System, Atmosphere, 12, https://doi.org/10.3390/atmos12060723, 2021

**(8) Figure quality and captions would strongly benefit from a thorough revision, such as**

**- Conistent use of (a) (b) etc. instead of top/middle left/right that is also sometimes used**

**- Explanation of abbreviations and variable names in caption**

**- Same chronological order of ERA reanalysis products (currently ERA5-Land sometimes comes last, sometimes first in figures).**

**- Consistent use of variable names.**

**For related and other comments, see annotated pdf.**

Thanks, we have tried to make figures more consistent by following all reviewer recommendations.

**(9) The lake temperature skill of ERA5-Land depended a lot on the lake depth information. The evaluation was thus stratified by whether lake depth improved (got more realistic) in ERA5-Land compared to ERA5. It's unclear on which information this stratification was made. How can a higher spatial resolution can lead to less realistic lake depth in ERA5-Land?**

The current lake mean depth file used at ECMWF is based on the Global Lake Database (See Table S1 of the original manuscript), and it has native resolution of 1 km. It consists of more than 14,000 in situ lake mean depth observations mapped on to a gridded file, and of indirectly estimated depths for all the rest of lakes based on a geomorphological approach. This means that the mean depth for some lakes which are not common for certain geomorphological regions might be estimated incorrectly – i.e., they can be overestimated or underestimated. Once lakes depths are aggregated to a coarser resolution, the depth for uncommon lakes can change and even become closer to the observed value but for the wrong reason (due to the other lake depths in the vicinity).

**(10) Figure 15: the spread of the bias values for the Bowen ratio is strongly increased for ERA5-Land. Either it's wrongly calculated/shown here or requires clarification.**

The code to produce these plots has been reviewed and we can confirm there are no errors. Nonetheless, the calculation of the Bowen ratio is usually prone to error, since it results from the ratio between two fluxes which can be positive or negative (so it is numerically unconstrained). This may create differences when large pixels are compared to in situ measurements. To reduce the influence of outliers, fluxes are first aggregated to daily resolution before calculating their ratio, as recommended by Shuttleworth (2012). We acknowledge, in any case, that the spread of the bias for the Bowen ratio is a bit higher for ERA5-Land. This relates mainly to a few outliers, as the quantiles are quite similar (as shown by the dashed lines that actually match up when comparing ERA5 and ERA5-Land), and close to each other. We computed the number of outliers found that are either "< q25 - 1.5 * (q75 - q25)" or "> q25 + 1.5 * (q75 - q25)". For ERA5 we found 12 observations meetings these conditions, whereas for ERA5-Land we found 20. This is not a strict definition of 'outlier', but it is analogous to the way they were computed by Martens et al. (2020).

The two references in this response:

Shuttleworth, W.J. (2012). Surface Energy Fluxes. In Terrestrial Hydrometeorology, W.J. Shuttleworth (Ed.). https://doi.org/10.1002/9781119951933.ch4

Martens, B., Schumacher, D. L., Wouters, H., Muñoz Sabater, J., Verhoest, N. E. C., and Miralles, D. G.: Evaluating the land-surface energy1005partitioning in ERA5, Geosci. Model Dev., 13, 4159–4181, https://doi.org/10.5194/gmd-13-4159-2020, 2020.

**(11) Several more comments are included in the annotated pdf;**

**L36:** We have added two extra, more recent references to highlight the need for land model developments:

Vereecken, H., Weihermüller, L., Assouline, S., Šimůnek, J., Verhoef, A., Herbst, M., Archer, N., Mohanty, B., Montzka, C., Vanderborght, J., Balsamo, G., Bechtold, M., Boone, A., Chadburn, S., Cuntz, M., Decharme, B., Ducharne, A., Ek, M., Garrigues, S., Goergen, K., Ingwersen, J., Kollet, S., Lawrence, D.M., Li, Q., Or, D., Swenson, S., de Vrese, P., Walko, R., Wu, Y. and Xue, Y. (2019), Infiltration from the Pedon to Global Grid Scales: An Overview and Outlook for Land Surface Modeling. Vadose Zone Journal, 18: 1-53 180191. https://doi.org/10.2136/vzj2018.10.0191

Boussetta, S., Balsamo, G., Arduini, G., Dutra, E., McNorton, J., Choulga, M., Agustí-Panareda, A., Beljaars, A., Wedi, N., Muñoz Sabater,J., de Rosnay, P., Sandu, I., Hadade, I.,

Carver, G., Mazzetti, C., Prudhomme, C., Yamazaki, D., and Zsoter, E.: ECLand: The ECMWFLand Surface Modelling System, Atmosphere, 12, https://doi.org/10.3390/atmos12060723, 2021

**L38**: We believe the sentence is right. Atmospheric reanalysis are produced with a single version of the coupled atmosphere-ocean-land prediction system and therefore show, despite the analysis increments, temporal consistency over the production period. This doesn't occur with the forecast models (as the IFS), which every certain period of time are updated (parameterizations, assimilation system, etc.).

**L41:** done, thanks.

**L57:** We prefer to keep the wording "climate studies" since it includes in a broader sense several inter-related investigations such as "climate change studies", "climate modelling studies" or "climate trend studies".

**L86:** the type of grid has been added to the main manuscript, as well as a supporting reference: "ERA5-Land produces a total of 53 variables describing the water and energy cycles over land, globally, hourly and at a spatial resolution of 9 km, matching the ECMWF triangular-cubic-octahedral (TCo1279) operational grid (Malardel et al., 2016)".

And the reference added:

Malardel, S., Wedi, N., Deconinck, W., Diamantakis, M., Kuehnlein, C., Mozdzynski, G., Hamrud, M., and Smolarkiewicz, P.: A new grid for the IFS, ECMWF Newsletter, 146, 23–28, https://doi.org/10.21957/zwdu9u5i, 2016.

**L102:** coordinates swapped. Thanks.

**L103:** ERA5-Land has been added to the figure and now it is used consistently over all the manuscript.

**L117:** "[...], no prior long ERA5 stream was available." has been replaced by " [...], a long, prior ERA5 stream was not available". For consistency the "prior long" in Line 110 has also benn replaced by "long, prior".

**Figure 3** → the figure has been improved as suggested by the reviewer.

**L118:** The following sentence "[...], and then letting three spin-up years." has been replaced by ", and then allowing the system to spin-up for three years".

**L120:** for the 3$^{rd}$ stream only one spin-up year was feasible (1949) because there wasn't ERA5 forcing available prior to Jan 1949. The available climatology (1981-2010) was used to initialize the spin-up year. This was indicated in Line 120 of the original manuscript.

**L127:** "approximately 10 m" → It is indeed 10 m exactly. The text has been changed to "[...], which is 10 m above the surface [...]

**L133:** "conventional" refers to all conventional meteorological observations (ships, radiosondes, SYNOP stations, etc.). We have added the word "conventional meteorological" to the main text.

**L151: Fig.4** → Following the reviewer recommendation, we have included the location of the three mentioned lakes in the top-right subplot of Fig.4, and updated the caption of the figure accordingly.

**L162: Specify which carbon fluxes are estimated. Is heterotrophic respiration simulated? Are longer term carbon stock changes estimated?**

The carbon fluxes are the Net ecosystem exchange (NEE), the Gross Primary production (GPP) and the Ecosystem respiration (Reco). Long term carbon stock is not estimated, instead ecosystem respiration is parametrized in an NWP adapted way as a function of land-use and soil temperature. The type of carbon fluxes has been added in the main text, lines 164-165 of the revised manuscript.

**Fig4.-->** We have replaced ERA5L by ERA5-Land in the title of each subplot. We have also defined LST in the figure caption.

**L173: Which crop? Constant parameters? I guess it's not seasonally changing and does not include harvest and bare soil period?**

There is no sub-type of crops (such C3 OR C4) considered in the actual version, it is the crops and mixed farming type from the BATS classification (table 8.1 of the IFS documentation). It is parametrised with a constant canopy resistance with no consideration of seasonal change or harvest.

**L183:** Yes, thanks. Added in parenthesis "the latter also a component of the water cycle".

**L213:** "OI" replaced by "Optimal Interpolation".

**L235:** Thanks, "averaged" replaced by "average".

**Eq.1:** Thanks for this comment. Indeed, the reviewer is right that the results will be equivalent either normalizing or not by the standard deviation. However, the standard definition of anomaly correlation for soil moisture is as displayed in eq.1, and for consistency with many other paper in this respect we prefer to keep it as it is.

**L240-241: STDD:** It is indeed equivalent to ubRMSE. We have added to the main text for clarification.

**L327:** Thanks to point this out. Although the reviewer is technically correct, the equation is a direct substitution of terms into the general skill score equation, so we believe it is clearer to leave it as it is in the text.

**L340: gaps:** The data downloaded from FLUXNET is gap-filled using different methods, each of them with some uncertainty. They also include a flag to indicate which method was used. If there was gap-filling in the time series (the flag is set) it simply means there was no measurement at that time. If it is gap-filled these data were not used as validation data, because there is some uncertain algorithm behind it. Therefore, a 'gap' in the text means any record that was gap-filled or has an NaN (that could not be gap-filled). We simply masked out those gap-filled records to retain only the actual measurements from the eddy-covariance sites.

We agree that the sentence in the main text is quite confusing, so we have removed the sentence in Line 346 of the original manuscript: "After quality control of in situ stations, only records without gaps during the whole period 2001-2014 were retained.", and we have replaced in lines 345-346: "(2) the removal of gap-filled records." by "(2) the removal of gap-filled records to retain only the actual measurements from the eddy-covariance sites".

**Energy Balance Closure:** The reviewer is quite right pointing this out. Indeed, we find that while some authors have applied energy balance closure, many others haven't done it. The reason is that there isn't clear accepted guidelines on how to do it properly, and in addition all the fluxes measured (including ground-heat flux) are needed to this end. Not every EC tower measures all the fluxes (ground heat flux is for instance often missing) which means that the sample of towers we can use for validation would drop drastically.

In order to make this point clear, we have added the following sentence in the main text, lines 348-352 of the revised manuscript: "Note that the measured energy fluxes used as reference in this paper were not corrected for energy balance closure because the number of towers used for validation would be drastically reduced, as the ground-heat flux is also needed and is not available from many towers. Some authors have already highlighted the lack of closure in the energy balance at eddy-covariance sites and a consequential tendency to underestimate the latent heat flux (Wilson et al., 2002, Ershadi et al., 2014, Jimenez et al., 2018)."

The added references are:

Wilson, K., Goldstein, A., Falge, E., Aubinet, M., Baldocchi, D., Berbigier, P., Bernhofer, C., Ceulemans, R., Dolman, H., Field, C., Grelle, A., Ibrom, A., Law, B., Kowalski, A., Meyers, T., Moncrieff, J., Monson, R., Oechel, W., Tenhunen, J., Valentini, R., and Verma, S.: Energy balance closure at FLUXNET sites, Agric. Forest Meteorol., 113, 223–243, doi:10.1016/S0168-1923(02)00109-0, 2002.

Ershadi, A., McCabe, M., Evans, J., Chaney, N., and Wood, E.: Multi-site evaluation of terrestrial evaporation models using FLUXNET data, Agric. Forest Meteorol., 187, 46–61, https://doi.org/https://doi.org/10.1016/j.agrformet.2013.11.008, 2014.

Jiménez, C., Martens, B., Miralles, D. M., Fisher, J. B., Beck, H. E., and Fernández-Prieto, D.: Exploring the merging of the global land evap-955oration WACMOS-ET products based on local tower measurements, Hydrol. Earth Syst. Sc., 22, 4513–4533, https://doi.org/10.5194/hess-22-4513-2018, 2018.

**L365:** "at a daily time scales" replaced by "at daily time scales".

**L409:** Thanks to spot the error, "represents" replaced by "represent".

**L417:** "m a.s.l." added here and elsewhere in the main text

**Fig.9** caption updated as suggested.

**L448:** the Mean Absolute Error has been added to the text on top of the percent.

**L456:** The statistical significance was tested with the Kruskal-Wallis test by ranks (best for non-Gaussian distribution variables), as written in line 287 of the original manuscript. We have deleted the word "modest".

**Fig.12 caption:** We have removed the gradations and added the degree symbol, and for consistency we have done the same for Fig.11.

**Fig.13.** Fig. 13 summarises the distribution of scores for 1285 river discharge stations as a CDF for both models (i.e GloFAS-ERA5 and GloFAS-ERA5-Land). This is a valid and commonly used method for hydrological model performance comparison (e.g. Alfieri et al., 2020; Kratzert et al., 2019). The reason why panel (a) is larger is that it is the main result for the overall KGE' metric. Because the KGE' can be decomposed into three components (i.e. correlation, bias ratio and variability ratio), we added this additional information as three secondary panels (i.e. (b), (c) and (d)), to help diagnose the patterns seen in the summary KGE' in (a).

The two references in this response are:

- Alfieri, L., Lorini, V., Hirpa, F. A., Harrigan, S., Zsoter, E., Prudhomme, C., and Salamon, P.: A global streamflow reanalysis for 1980–2018, Journal of Hydrology X, 6, 100049, https://doi.org/10.1016/j.hydroa.2019.100049, 2020.

- Kratzert, F., Klotz, D., Shalev, G., Klambauer, G., Hochreiter, S., and Nearing, G.: Towards learning universal, regional, and local hydrological behaviors via machine learning applied to large-sample datasets, 23, 5089–5110, https://doi.org/10.5194/hess-23-5089-2019, 2019.

**L483-484:** Latter in line 484 it is explained that the bias show (observation – model). Anyway, following the reviewer recommendation we have moved the description of bias (and other metrics used for the energy fluxes) to section 3.7.1, and made this sentence clearer: "The left panel of Fig 15 shows the violin plots of the turbulent fluxes against in situ eddy-covariance measurements for ERA-Interim and ERA5-Land" has been replaced by "The left panel of Fig 15 shows the violin plots of the ERA-Interim and ERA5-Land turbulent fluxes (compared to in situ eddy-covariance measurements)"

**L485: anomaly correlation:** We use the anomaly correlation as the temporal variability of the turbulent fluxes is strongly influenced by the seasonal cycle of its main drivers at the scales considered in this experiment, and therefore the performance of the land-surface schemes in response to anomalous weather conditions (i.e. with respect to the seasonal cycle) might be masked when raw time series are analysed. We have also used the standardised anomalies to allow us to directly compare the quality of the turbulent fluxes and the Bowen ration, despite their different order of magnitude. This was already explained in lines 346-349 of the original manuscript.

**Fig. 15:** Thanks; "Unlikely most metrics and variables" has been removed from the caption, making it more clear now.

We have tried to increase the font of the labels. However, since this is a side-by-side figure with two subplots, by increasing the font size the violin plots look smaller, so we decided to keep the current font size. However, we have moved the titles on top as recommended by this reviewer.

**L491:** We agree with the reviewer, and to make it more consistent through the main text, we have replaced all "sshf and slhf" entries by "H" and "λρE", respectively.

**L498-499:** We already provided a plausible explanation in the "conclusions and discussion" section, in lines 617-619 of the original manuscript.

**Fig.16:** To generate this figure we used a projected CRS, which allows to show the background map. We have added the X (Easting) and Y (Northing) coordinates to the figure in EPSG3857 (https://epsg.io/3857), but not lat-lon as they just would be approximate. The result is the following:

[Figure]

We believe that they look more confusing than before, and the background map provides more context on the location of the sites than simple lat/lon coordinates. So we decided to leave the figure as it was shown originally.

**L504:** Please, see response to comment (10)

**L506:** We have revised this sentence and we have removed "and for the stations where the recorded air temperature is milder", as the figure is indeed not conclusive for air temperatures.

**L512:** Thanks to spot the misspelling, "term" has been replaced by "terms".

**L515-516**: Just to clarify: GLEAM+ERA5 is forced with T and R from ERA5, whereas GLEAM+ERA5-Land is forced with T and R from ERA5-Land, which is described in Section 3.7.2. However we agree that this sentence is confusing and therefore we have decided to remove it from the main text.

**Fig.19**: As recommended. by the reviewer, the ranges have been adjusted for subplots c) and d). And indeed the improvement from ERA5-Land to ERA-Interim in terms of correlation values is now much more clear. We have also increased the resolution of the figure. In addition and to be consistent, we have updated the figure S5 of the supplementary material.

**Fig.19 caption.** Thanks to spot the error, indeed the figure refers to ERA5-Land and not to ERA5. Corrected. In addition, we have rephrased the whole caption as follows:

"Global statistics of ERA5-Land (left column) and ERA-Interim (right column) Land Surface Temperature (LST, in K) for the time period 2003-2018. The first row are the global maps of mean LST, the second row are the correlation maps and third row are the RMSE maps, with respect to the MODIS LST average ensemble. Note that the LST here refers to the skin temperature."

**Fig.20:** Each triangle represents a selected pixel, but the colour hasn't got any other meaning.

The word "quantitative" has also been removed from the caption as suggested.

**L552:** As recommended by the reviewer, we have improved the wording of this sentence:

"Overall, the water cycle is improved in ERA5-Land compared to ERA5 according to the different variables evaluated, whereas the energy cycle variables show similar performance; both ERA5 and ERA5-Land perform substantially better than ERA-Interim. The main evaluation findings are as follows":

**L554:** Here we didn't use significance in a statistical sense. To clarify this point we have rephrased the whole sentence in Line 568-569 of the revised manuscript:

"Significant improved correlation of the ERA5-Land (compared to ERA-Interim and ERA5) root-zone soil moisture absolute and anomaly values with respect to in situ measurements is achieved for almost all networks assessed" has been replaced by:

"Soil moisture: the boxplots show a consistent improvement of the statistical metric distributions of ERA5-Land with respect to those of ERA5, in particular the improvement is more marked for the root-zone soil moisture."

**L562:** The sentence is about the minimum saturation level linked to the soil type and in the top layer both evaporation and transpiration would occur so we prefer to keep the term evapotranspiration

**L590:** By "lakes with more realistic depths in ERA5-Land" we mean taking into account, in the statistics, only lakes whose depth is closer to the observations using the ERA5-Land spatial grid, and not taking into account lakes whose depth remained unchanged either in ERA5 or in the ERA5-Land grid. This is because when spatial resolution changes some lake depths become closer to the in situ observations, some further from in situ, and some lake depths remain the same - for instance because it is a big lake without bathymetry. In order to make this sentence more clear, we have replaced:

"Isolating lakes with more realistic depths in ERA5-Land reduces […]" by "Isolating lakes whose depth is more realistic using the ERA5-Land grid (i.e. only lakes whose depth in the ERA5-Land grid match better in situ observations) the LSWT errors are reduced by more than 20%".

See also our comment to main point (9) of this reviewer.

**L595-599:** We apologize for the confusion created, as the sentences refer to different validation datasets (Alqueva, Finish lakes or the global inventory). We have rephrased lines 589-600 of the original manuscript to make these conclusions more clear:

"[...]. The higher resolution of ERA5-Land allows, in many cases, a more accurate specification of lake depth. Thus, isolating lakes whose depth is more realistic using the ERA5-Land grid (i.e. only lakes whose depth in the ERA5-Land grid matches better in situ observations), the LSWT MAE is reduced by more than 20\%. The positive influence of a higher resolution atmospheric forcing was also verified by isolating lakes whose depth remained unchanged either using ERA5 or ERA5-Land.

Both reanalysis LSWT estimates were also compared using as reference a global inventory (1995-2009) based on summer observations from satellite sensors. In this case, the performance of both reanalysis is quite similar (excluding exceptional lakes where Flake performs poorly). The uncertainty of lake depths in the global inventory is larger and it could have an impact on these results. However, even when comparing only lakes with a depth less than 50 m (Flake was designed for medium-depth lakes) the performance of ERA5-Land LSWT does not show significant improvement compared to ERA5 (based on MAE). This might be due to the averaging technique used for the in situ measurements, which uses satellite measurements representing one instant in time rather than continuous hourly data. Nevertheless, in summer months, ERA5 LSWT biases are on average 2.2°C cooler than observations, whereas ERA5-Land LSWT is just 1.3°C cooler."

**L600:** Thanks to point this out. Due to the high resolution grid of ERA5-Land, the mean depth of many lakes are more realistic in ERA5-Land, what it implies that the total volume is improved as well. So we believe that the improvement of the lake characterization also helps to improve the river discharge at global scale. We have added the word "[...] and lakes characterization [...]"

**L610:** In the Amazon basin, the bias term is a little bit lower in ERA5-land (so bit less water), but it is really a subtle difference. Being ERA5 precipitation a forcing parameter of ERA5-Land, we assume similar precipitation values in both reanalysis, so an overestimation of evaporation remains the most likely reason. Since the differences are, for most discharge stations, very small, we really didn't undertake a deep diagnostic though.

Consequently, we have replaced "The degradation in the Amazon is small, but it is linked to an underestimation of the surface and subsurface runoff (not shown)." by "The degradation in the Amazon is small, linked to an underestimation of the surface and subsurface runoff (not shown) that is likely caused by a slight overestimation of evaporation."

**L639:** We have replaced "radiative aspect" by "surface fluxes". Thanks.

**L641:** "land surface" replaced by "land surface modeling"

**L642:** We agree that a trend analysis would be beneficial, but it is out of the scope of this paper and it would require another dedicated paper only for this aspect. See also extended responses to comment (1) of reviewer 1 and 2.

**L646:** "international" replaced by "internationally".

**L656:** ";" replaced by "."

**L677:** Please see response to comment in line 173

**L680-681:** The canopy resistance for water vapour is computed with the JARVIS approach, then an A-gs based formulation is derived to compute the carbon fluxes. This modular approach is adopted in order not to deteriorate the NWP scores which are calibrated based on the JARVIS formulation. Future developments will focus on a more explicit and tight coupling of the carbon and water fluxes.

**L711:** We are actually exploring the improvement of both the discretisation of the soil layer as well as the soil parameters that the reviewer refers to. We have modified this paragraph to account for other ongoing developments (see response to main point 7 of this reviewer).

---

## Author Response (AR2)

Dear reviewer,

Thank you for your further comments that we have taken into account:

**Authors should still improve presentation quality.**
**Only giving examples due to lack of time:**
**Figure 2, line break precipitation**

Corrected. This occurred when converting to jpg, and now solved by using a png file

**Figure 3, unit style**

Unfortunately the software used to add the unit does not allow to change the style.

**Figure 15, Mean Absolute Error (MAE) is supposed to have a unit**

The figure shows 'standardized' MAE, and therefore without units. We have added it in the legend and make it consistent through all the document.

We still believe Table 1 and Table A2 need to be improved. We didn't find a quick fix for them in latex, as tomorrow is the deadline to re-submit this paper. However, during the editorial process we hope to have another chance to provide a better quality table. For the rest of the document, we believe the aspect-ratio of some figures could be better, but again we hope to make it better during the editorial process.

Kind regards,

Joaquin Munoz Sabater